# Learning from positive and unlabeled examples
# -Finite size sample bounds

**Farnam Mansouri**
University of Waterloo and Vector Institute
`f5mansou@uwaterloo.ca`

**Shai Ben-David**
University of Waterloo and Vector Institute
`shai@uwaterloo.ca`

## Abstract

PU (Positive Unlabeled) learning is a variant of supervised classification learning in which the only labels revealed to the learner are of positively labeled instances. PU learning arises in many real-world applications. Most existing work relies on the simplifying assumptions that the positively labeled training data is drawn from the restriction of the data generating distribution to positively labeled instances and/or that the proportion of positively labeled points (a.k.a. the class prior) is known apriori to the learner. This paper provides a theoretical analysis of the statistical complexity of PU learning under a wider range of setups. Unlike most prior work, our study does not assume that the class prior is known to the learner. We prove upper and lower bounds on the required sample sizes (of both the positively labeled and the unlabeled samples).

## 1 Introduction

Learning from positive and unlabeled data (PU learning) is a variant of binary classification prediction semi-supervised learning, where the training data consist only of positively labeled and unlabeled examples. PU learning arises in many applications, such as personal advertisement (where a person is labeled according to whether a given add is relevant to them. When a person responds to the add, we know they belong to the set of positive instances. However, we cannot tell the label of unresponsive customers), land cover classification Li et al. (2010) (say, we wish to classify forest land cover from aerial images, where training data consist of unlabeled land images and forest aerial images), prediction of protein similarity Elkan and Noto (2008) and many other applications like knowledge base completion Bekker and Davis (2020), disease-gene identification Yang et al. (2012) and more.

Standard machine learning paradigms, such as empirical risk minimization (namely, training a classifier to minimize the miss-classification loss over the training data) or regularized risk minimization may fail badly in such settings, since their success guarantees rely on having access to labels from both classes (positive and negative labels). We are interested in finite sample size generalization guarantees. Having a weaker supervision than standard fully supervised learning, achieving generalization bounds for PU learning requires stronger assumptions. In this work, we show how some of the common assumptions used in this domain can be relaxed, while also showing some negative, impossibility results.

**Various setups for PU learning.** We consider the case in which both training samples (the positively labeled and the unlabeled examples) are generated by random processes unknown to the learner. The learner's goal is to obtain a classifier that minimizes misclassification with respect to *a target evaluation distribution* $\mathcal{D}$ over $\mathcal{X} \times \{0, 1\}$ (where $\mathcal{X}$ is the domain set). Training data generating setup can be viewed along two basic axes; The first is whether the positively labeled data is generated independently of the unlabeled sample (as opposed to the case where the positively labeled examples

39th Conference on Neural Information Processing Systems (NeurIPS 2025).

are sampled from an already sampled unlabeled set of instances). The other axis is the labeling mechanism through which positive labels are assigned to training examples.

Our paper focuses on scenarios where the unlabeled sample and the positive sample are independent of each other (called *case-control scenarios* by Niu et al. (2016)). As an example of a case-control scenario, consider the task of predicting whether a given profile will become a user of a mobile application. For this example, the positive sample can be collected from individuals who are already users of the application, while the unlabeled sample can be drawn from a broader pool of random individuals.

Let $\mathcal{D}_+$ and $\mathcal{D}_-$ denote the conditioning of $\mathcal{D}$ on the label being positive or negative respectively. We consider four setups for how positive training data is generated (See Section 4 for formal definitions):

- *Selected completely at random (SCAR)* Elkan and Noto (2008): Positive training data is drawn i.i.d. from $\mathcal{D}_+$.
- *Selected at random (SAR)* Bekker et al. (2019): Positive training data are drawn i.i.d. from a distribution whose support is a subset of the support of $\mathcal{D}_+$.
- *Positive covariate shift (PCS):* Positive training data is drawn from a distribution that shares the same labeling function as $\mathcal{D}$ but has different marginal distributions (referred to as *positive-only shift* in Sakai and Shimizu (2019)).
- *Arbitrary positive distribution shift (APDS):* Positive training data is drawn from an arbitrary distribution (generalization bounds in this case depend on measures of similarity between the two distributions).

Following the common terminology, *realizable* setup refers to learning with respect to data distributions for which some member of the concept class has zero misclassification loss. The setup is *agnostic* PU learning when no such condition is assumed. The *class prior* is the probability of positive labels, $\alpha := \mathcal{D}[\{(x,1) : x \in \mathcal{X}\}]$. In most prior work on PU learning, the class prior is assumed as prior knowledge.

**Our Contributions.** The main high-level contributions of this paper are as follows:

1. We provide finite sample complexity bounds without relying on knowledge of the class prior $\alpha$. To the best of our knowledge, prior provable results in PU learning typically assume that $\alpha$ is known and used by the learner. The only exceptions are Liu et al. (2002), which provides a result limited to realizable PU learning under the SCAR setup, Lee et al. (2025) that are studying a setup where unlabeled data is sampled from a distribution different from the target evaluation data, and Kato and Teshima (2021); Zheng et al. (2022) which study specific classes of neural networks.

2. We provide new sample complexity upper bounds in a variety of setups, for which such bounds have not been previously proved.

3. We prove novel lower bounds that match existing positive results for the SCAR setup.

In more detail, our contributions are:

- *Realizable PU Learning (SCAR setup).* In Theorem 1, we provide a lower bound on the sample complexity of positive examples that nearly matches earlier upper bounds (e.g., by Liu et al. (2002)). Moreover, in Theorem 3, we also provide a lower bound on the sample complexity of unlabeled examples based on a novel combinatorial parameter that we introduce, called *claw number*.
- *Realizable PU learning (SAR setup).* We prove the first finite sample complexity for this setup that does not require knowledge of $\alpha$ (Theorem 8). We then provide an almost tight lower bound on the sample complexity of positive examples in Theorem 9.
- *Realizable PU learning (PCS setup).* For this setup, we introduce the first algorithm which guarantees finite sample complexity (Theorem 12). We then provide lower bounds on sum of sample complexity of positive and unlabeled examples (Theorem 10 and Theorem 11). These results highlight the differences between the PCS and the SAR setups.
- *Agnostic PU Learning (SCAR setup, when $\alpha$ is known).* For this setup, in Theorem 13, we propose a lower bound on the sample complexity of both positive and unlabeled examples that nearly matches existing upper bounds established in Du Plessis et al. (2015).

- *Agnostic PU Learning (SCAR setup, when $\alpha$ is unknown).* While without knowledge of $\alpha$ or additional assumptions on the data or concept class it is impossible to find a classifier whose misclassification rate is arbitrarily close to that of the best in the class, we show in Corollary 18, that a learner can always find a classifier whose misclassification rate is arbitrarily close to $\frac{\max(\alpha, 1-\alpha)}{\min(\alpha, 1-\alpha)}$ times the misclassification rate of the best concept in the concept class. Moreover, in Corollary 14, we show that this multiplicative factor is tight. Our result also yields an improved generalization bound in scenarios where an approximation of $\alpha$ is available.
- *Agnostic PU learning (APDS setup).* For this setup, we derive the first generalization bounds with finite sample complexity (Theorem 13).

Table 1: Summary of all results presented in this paper (excluding those in Section 5.2). Here, $d$ denotes the VC-dimension of the concept class $\mathcal{C}$ ; $k$ is the dimensionality of the input space; $\mathfrak{h}$ is the claw number of $\mathcal{C}$; $r$ is a weight ratio between distribution of positively labeled training data and $\mathcal{D}_+$ ; $\gamma$ is the margin parameter ;$\pi$ is any lower bound on $\alpha$ that is available to the learner. Logarithmic factors are suppressed in this table.

| Bounds on the sample complexity of PU learning | | | |
|---|---|---|---|
| Realizable (SCAR) | $m_{\mathcal{C}}^{pos}(\varepsilon, \delta) = \tilde{O}\left(\frac{d}{\varepsilon}\right)$ | $m_{\mathcal{C}}^{unlabel}(\varepsilon, \delta) = \tilde{O}\left(\frac{d}{\varepsilon}\right)$ | Liu et al. (2002) |
| | $m_{\mathcal{C}}^{pos}(\varepsilon, \delta) = \tilde{\Omega}\left(\frac{d}{\varepsilon}\right)$ | $m_{\mathcal{C}}^{unlabel}(\varepsilon, \delta) = \tilde{\Omega}\left(\frac{\mathfrak{h}}{\varepsilon}\right)$ | **New results in this work.** |
| Realizable (SAR) | $m_{\mathcal{C}}^{pos}(\varepsilon, \delta) = \tilde{O}\left(\frac{d}{r\varepsilon}\right)$ | $m_{\mathcal{C}}^{unlabel}(\varepsilon, \delta) = \tilde{O}\left(\frac{d}{\varepsilon}\right)$ | **New results in this work.** |
| | $m_{\mathcal{C}}^{pos}(\varepsilon, \delta) = \tilde{\Omega}\left(\frac{d}{r\varepsilon}\right)$ | $m_{\mathcal{C}}^{unlabel}(\varepsilon, \delta) = \tilde{\Omega}\left(\frac{\mathfrak{h}}{\varepsilon}\right)$ | **New results in this work.** |
| Realizable (PCS) | $m_{\mathcal{C}}^{pos}(\varepsilon, \delta) = \tilde{O}\left(\frac{d}{r^2\varepsilon}\right)$ $\quad m_{\mathcal{C}}^{unlabel}(\varepsilon, \delta) = \tilde{O}\left(\frac{\left(\frac{\sqrt{k}}{\gamma}\right)^k + \pi d}{\pi\varepsilon}\right)$ | | **New results in this work.** |
| | $m_{\mathcal{C}}^{pos}(\varepsilon, \delta) + m_{\mathcal{C}}^{unlabel}(\varepsilon, \delta) = \tilde{\Omega}(1 + 1/2\gamma)^{k/2}$ | | **New results in this work.** |
| Agnostic (SCAR, known $\alpha$) | $m_{\mathcal{C}}^{pos}(\varepsilon, \delta) = \tilde{O}\left(\frac{d}{\varepsilon^2}\right)$ | $m_{\mathcal{C}}^{unlabel}(\varepsilon, \delta) = \tilde{O}\left(\frac{d}{\varepsilon^2}\right)$ | Du Plessis et al. (2015) |
| | $m_{\mathcal{C}}^{pos}(\varepsilon, \delta) = \tilde{\Omega}\left(\frac{d}{\varepsilon^2}\right)$ | $m_{\mathcal{C}}^{unlabel}(\varepsilon, \delta) = \tilde{\Omega}\left(\frac{d}{\varepsilon^2}\right)$ | **New results in this work.** |

**Related works.** We briefly survey previous theoretical studies of PU learning. We start with works on the SCAR setup. For the easiest case of realizable learning, Liu et al. (2002) describe an algorithm with finite sample complexity.

For the more challenging agnostic PU learning setting, previously proposed approaches typically rely on a priori knowledge of the class prior (e.g., Du Plessis et al. (2015)). When the class prior is unknown, existing studies often impose restrictive assumptions on the underlying distribution or the concept class. Much of this literature focuses on estimating the prior $\alpha$. The assumptions employed in these works include: (i) *Separability:* non-overlapping support between the $\mathcal{D}_-$ and $\mathcal{D}_+$ Elkan and Noto (2008); Du Plessis and Sugiyama (2014); (ii) *Anchor set*: requiring a subset of the instance space defined by partial attribute assignment, to be purely positive Scott (2015); Liu and Tao (2015); Christoffel et al. (2016); Bekker and Davis (2018); (iii) Ramaswamy et al. (2016) discuss a generalization of anchor set assumption and call it also separability; (iv) Irreducibility: $\mathcal{D}_-$ cannot be expressed as a linear combination of $\mathcal{D}_+$ and any other distribution Blanchard et al. (2010); Jain et al. (2016). There are also studies focusing on specific classes of neural networks Kato and Teshima (2021); Zheng et al. (2022), which adopt density-ratio estimation method. Our results do not rely on any of these assumptions.

Next, we consider studies of PU learning that extend beyond the SCAR setup. Several articles examine PU learning in the SAR setting Coudray et al. (2023); Dai et al. (2023); Gong et al. (2021); Na et al. (2020); Bekker et al. (2019); Kato et al. (2019); He et al. (2018), among which only Coudray et al. (2023); Gong et al. (2021); Kato et al. (2019); He et al. (2018) pursues theoretical analysis (the others focus primarily on empirical evaluations). In contrast to our work, these studies assume that $\alpha$ is known. Under that assumption, they provide learnability results applicable to the agnostic PU learning setting. Kato et al. (2019) focuses on establishing statistical consistency rather than finite sample guarantees. He et al. (2018) analyze a special case of the SAR setting, referred to as the *probabilistic gap assumption*.

Sakai and Shimizu (2019); Hammoudeh and Lowd (2020); Kumar and Lambert (2023) discuss statistical consistency for different variations of PU learning. This is in contrast with our focus on finite sample size generalization bounds. Lee et al. (2025) studies both the sample complexity and

computational complexity in a setting where the distribution of unlabeled training data is drawn from a distribution that can differ from the target evaluation distribution, while the positive training data is drawn from the target evaluation distribution conditioned on the label being positive.

Note that, due to space constraints, all proofs in this submission are deferred to the appendix.

## 2 Setting

We consider the following setup for learning with positive and unlabeled examples (PU learning). Let $\mathcal{X}$ be the domain set, $\mathcal{Y} = \{0, 1\}$ the labels set. We consider two distributions, a distribution $\mathcal{D}$ over $\mathcal{X} \times \mathcal{Y}$, and a distribution for *sampling the positively labeled training data* over $\mathcal{X}$ denoted by $\mathcal{P}$. Given a function $f : \mathcal{X} \to \mathcal{Y}$, define $\mathrm{err}_{\mathcal{D}}(f) := Pr_{(x,y)\sim\mathcal{D}}[f(x) \neq y]$ as the error of $f$ with respect to $\mathcal{D}$. Also, define *positive distribution*, and *negative distribution* respectively to be $\mathcal{D}_+(A) := \mathcal{D}(A \mid y = 1)$ and $\mathcal{D}_-(A) := \mathcal{D}(A \mid y = 0)$ for every measurable set $A \subseteq \mathcal{X}$. Moreover, denote $\mathcal{D}_{\mathcal{X}}$ to be the marginal distribution of $\mathcal{D}$ over the domain set.

A PU learner takes (i) a sample $S^U$ of size $a$ i.i.d. drawn from marginal distribution $\mathcal{D}_{\mathcal{X}}$, and (ii) a sample $S^P$ of size $b$ i.i.d. drawn from $\mathcal{P}$ independent of $S^U$, denoted by $S^P$, and similar to classical machine learning, it aims to output a function $f : \mathcal{X} \to \mathcal{Y}$ which minimizes $\mathrm{err}_{\mathcal{D}}(f)$. Formally, a PU learner is a function

$$\mathcal{A} : \mathcal{X}^* \times \mathcal{X}^* \to \{0, 1\}^{\mathcal{X}}.$$

We now establish our framework for evaluating the success of PU learners:

**Definition 1** (PU learnability). *Let $\mathcal{C}$ be a concept class over domain $\mathcal{X}$. Moreover, let $\mathcal{W}$ be a set of pairs $(\mathcal{D}, \mathcal{P})$, where $\mathcal{D}$ is a distribution over $\mathcal{X} \times \mathcal{Y}$; and $\mathcal{P}$ is a distribution over $\mathcal{X}$. We say that concept class $\mathcal{C}$ is PU learnable over the class $\mathcal{W}$ if there exist functions $m_{\mathcal{C}}^{pos} : (0, 1) \times (0, 1) \to \mathbb{N}, m_{\mathcal{C}}^{unlab} : (0, 1) \times (0, 1) \to \mathbb{N}$, and a PU learner $\mathcal{A}$ such that for all $(\varepsilon, \delta) \in (0, 1) \times (0, 1)$ and distributions $(\mathcal{D}, \mathcal{P}) \in \mathcal{W}$ if $b > m_{\mathcal{C}}^{pos}(\varepsilon, \delta)$ and $a > m_{\mathcal{C}}^{unlab}(\varepsilon, \delta)$, we have*

$$\Pr_{S^P \sim \mathcal{P}^b, S^U \sim \mathcal{D}_{\mathcal{X}}^a} \left[ \mathrm{err}_{\mathcal{D}} \left( \mathcal{A}(S^P, S^U) \right) \geq \min_{c \in \mathcal{C}} \mathrm{err}_{\mathcal{D}}(c) + \varepsilon \right] < \delta.$$

*We also say $\mathcal{A}$ PU learns $\mathcal{C}$ over $\mathcal{W}$.*

**Notations:** Given any set $J$ and $k \in \mathbb{N}$, let $U_J$ denote the uniform distribution over $J$, and define $J^k := \{(j_1, \ldots, j_k) \mid j_1, \ldots, j_k \in J\}$, and $[k] := \{1, \ldots, k\}$. Given a family of distributions $\mathcal{D}_{\omega}$ over $\mathcal{X} \times \{0, 1\}$, where $\omega$ ranges over some parameter set, we respectively denote the marginal over $\mathcal{X}$, the positive distribution, and the negative distribution of $\mathcal{D}_{\omega}$ by $\mathcal{D}_{\mathcal{X}, \omega}, \mathcal{D}_{+, \omega}$, and $\mathcal{D}_{-, \omega}$.

Let $\mathcal{C}$ be a concept class over $\mathcal{X}$. Define $\mathcal{C}\Delta\mathcal{C} := \{c \oplus c' \mid c, c' \in \mathcal{C}\}$. Moreover, denote the best classifier as $c^* := \arg\min_{c \in \mathcal{C}} \mathrm{err}_{\mathcal{D}}(c)$, and $\min_{c \in \mathcal{C}} \mathrm{err}_{\mathcal{D}}(c)$ as the *approximation error*. Furthermore, for a function $c : \mathcal{X} \to \{0, 1\}$ and distribution $\mathcal{D}$ over $\mathcal{X} \times \{0, 1\}$, define the *false positive rate* as $\mathrm{err}_{\mathcal{D}}^+(c) := \Pr_{x \sim \mathcal{D}_+}[c(x) \neq 1]$, and the *false negative rate* as $\mathrm{err}_{\mathcal{D}}^-(c) := \Pr_{x \sim \mathcal{D}_-}[c(x) \neq 0]$. Given a subset $B \subseteq \mathcal{X}$, define $\mathcal{C} \cap B := \{c \cap B \mid c \in \mathcal{C}\}$. Moreover, for a multiset $S = (x_1, x_2, \ldots, x_m) \in \mathcal{X}^*$, define $\mathrm{Domain}(S) := \{x \mid x \in S\}$. Define the *restriction of $S$ to $B$* denoted by $S \mid B$ as the subsequence of elements $x_i \in S$ such that $x_i \in B$.

## 3 Analysis of Realizable PU Learning –SCAR setup

In this section, we study PU learning under the realizability assumption in the SCAR setup. It is already known that every concept class with finite VC dimension is PU learnable in this setting Liu et al. (2002). We begin by establishing lower bounds on the sample complexity. In particular, we provide a lower bound on the sample complexity of positive examples that nearly matches the upper bound established by Liu et al. (2002).

**Theorem 1.** *Let $\mathcal{C}$ be a concept class with VC dimension $d \geq 2$ over the domain $\mathcal{X}$. There exists a $M > 1$ such that for any number of positive samples upper bounded by $b \leq M \left( \frac{d + \ln(1/\delta)}{\varepsilon} \right)$ and for every number of unlabeled samples $a \in \mathbb{N}$, $(\varepsilon, \delta) \in (0, 1) \times (0, 1)$, and PU learner $\mathcal{A}$ there is a distribution $\mathcal{D}$ realized by $\mathcal{C}$ over $\mathcal{X} \times \{0, 1\}$ such that $\Pr_{S^P \sim \mathcal{D}_+^b, S^U \sim \mathcal{D}_{\mathcal{X}}^a} \left[ \mathrm{err}_{\mathcal{D}}(\mathcal{A}(S^P, S^U) \geq \varepsilon \right] > \delta.*

Next, we provide a lower bound for the sample complexity of unlabeled samples with respect to a combinatorial parameter we call *claw number*. Claw number is formally defined in the following. As mentioned in Remark 2, claw number is always smaller than VC dimension.

**Definition 2.** *Let $\mathcal{C}$ be a concept class over domain $\mathcal{X}$. We define* claw number *of $\mathcal{C}$ to be the largest $\mathfrak{h} \in \mathbb{N}$ such that for every $m \geq \mathfrak{h}$, there exists a $B \subseteq \mathcal{X}$ with $|B| = m$ such that $\{O \subseteq B \mid |O| = m - \mathfrak{h}\} \subseteq \mathcal{C} \mid B$. If no such $\mathfrak{h}$ exists, we say the claw number of $\mathcal{C}$ is 0.*

**Remark 2.** *Claw number of a class is always less than or equal to VC dimension. This is because for every $B \subseteq \mathcal{X}$ with $|B| \geq 2\mathfrak{h}$ we have $\mathrm{VCD}(\{O \subseteq B \mid |O| = |B| - \mathfrak{h}\}) \geq \mathfrak{h}$.*

**Theorem 3.** *Let $\mathcal{C}$ be a concept class with claw number $\mathfrak{h} \geq 1$. There exists a $M > 1$ such that for any number of unlabeled samples upper bounded by $a \leq M\left(\frac{\mathfrak{h}+\ln(1/\delta)}{\varepsilon}\right)$ and any number of positive samples $b \in \mathbb{N}$, $(\varepsilon, \delta) \in (0,1) \times (0,1)$, and PU learner $\mathcal{A}$ there is a distribution $\mathcal{D}$ realized by $\mathcal{C}$ over $\mathcal{X} \times \{0,1\}$ such that $\Pr_{S^P \sim \mathcal{D}_+^b, S^U \sim \mathcal{D}_{\mathcal{X}}^a}\left[\mathrm{err}_{\mathcal{D}}(\mathcal{A}(S^P, S^U) \geq \varepsilon\right] > \delta$.*

Note that Lee et al. (2025) showed that no concept class $\mathcal{C}$ with $\mathrm{VCD}(\mathcal{C}_\cap) = \infty$ (where $\mathcal{C}_\cap$ is defined below) is realizably PU learnable in the SCAR setup without access to unlabeled examples. Note that $\mathrm{VCD}(\mathcal{C}_\cap)$ is also studied as the slicing dimension in Kivinen (1995) and as the 1-centered star number in Hanneke (2024). The following proposition shows that Theorem 3 extends the results of Lee et al. (2025) by demonstrating that not only do positive examples alone not suffice when $\mathrm{VCD}(\mathcal{C}_\cap) = \infty$, but there also exists a concrete lower bound on the number of required unlabeled examples.

**Proposition 4.** *For a concept class $\mathcal{C}$, let $\mathcal{C}_\cap := \left\{\bigcap_{c \in A} c \mid \text{finite } A \subseteq \mathcal{C}\right\}$. Then $\mathrm{VCD}(\mathcal{C}_\cap) = \infty$ if and only if the claw number of $\mathcal{C}$ is at least 1.*

We then restate the Theorem 1 of Liu et al. (2002) in Corollary 6, providing an alternative proof based on the notion of $\varepsilon$-nets, which we formally define below. Our proof also leads to new results for the SAR and PCS setups, presented in Section 4.

**Definition 3** ($\varepsilon-$net). *Let $\mathcal{X}$ be some domain, $\mathcal{B} \subseteq 2^{\mathcal{X}}$ a collection of subsets of $\mathcal{X}$ and $\mathcal{Q}_{\mathcal{X}}$ a distribution over $\mathcal{X}$. An $\varepsilon$-net for $\mathcal{W}$ with respect to $\mathcal{Q}_{\mathcal{X}}$ is a subset $N \subseteq \mathcal{X}$ that intersects every member of $\mathcal{B}$ that has $\mathcal{Q}_{\mathcal{X}}$-weight at least $\varepsilon$.*

Let us also elaborate on the learning algorithm Liu et al. (2002) introduced, appearing in (1). Note that (1) simply selects the concept with the fewest number of 1s over $S^U$ among all concepts consistent with $S^P$. In this sense, it can be seen as a counterpart to *empirical risk minimization* in the PU learning setting. We therefore refer to any concept returned by (1) as a *positive empirical risk minimizer* (PERM).

$$\mathrm{argmin}_{c \in \mathcal{C}, Domain(S^P) \subseteq c} |c \mid S^U| \tag{1}$$

**Lemma 5.** *Let $\mathcal{C}$ be a realizable concept class with VC dimension $d$ over domain $\mathcal{X}$. Let $S$ be a sample i.i.d. drawn from $\mathcal{D}_{\mathcal{X}}$ and $T \in \mathcal{X}^*$ be an $\varepsilon-$net for $\mathcal{C} \triangle \mathcal{C}$ on $\mathcal{D}_+$ such that $Domain(T) \subseteq c^\star$. Denote $c^{PU} := \mathrm{argmin}_{c \in \mathcal{C}, Domain(T) \subseteq c} |c \mid S|$. Then there exists a $M > 1$ such that if $|S| > M\left(\frac{d \ln(1/\varepsilon) + \ln(1/\delta)}{\varepsilon}\right)$, then with probability $1 - 2\delta$ we have $\mathrm{err}_{\mathcal{D}}(c^{PU}) \leq 14\varepsilon$.*

**Corollary 6.** *[Theorem 1 of Liu et al. (2002)] Let $\mathcal{C}$ be a concept class with VC dimension $d$ over the domain $\mathcal{X}$. Let $\mathcal{W}$ be a set of duos $(\mathcal{D}, \mathcal{D}_+)$ such that $\mathcal{D}$ is realized by $\mathcal{C}$. Then $\mathcal{C}$ is PU learnable over $\mathcal{W}$ with sample complexity $m_{\mathcal{C}}^{pos}(\varepsilon, \delta), m_{\mathcal{C}}^{unlabel}(\varepsilon, \delta) = O\left(\frac{d \ln(1/\varepsilon) + \ln(1/\delta)}{\varepsilon}\right)$.*

*Proof.* For a fixed constant $M$, as long as $b > M\left(\frac{\mathrm{VCD}(\mathcal{C} \triangle \mathcal{C}) \ln(1/\varepsilon) + \ln(1/\delta)}{\varepsilon}\right)$ we have that $S^P$ with probability $1 - \delta$ is an $\varepsilon-$net for $\mathcal{C} \triangle \mathcal{C}$ on $\mathcal{D}_+$ (e.g., see Haussler and Welzl (1987)). Since $\mathrm{VCD}(\mathcal{C} \triangle \mathcal{C}) \leq 2 \mathrm{VCD}(\mathcal{C}) + 1$ (it can be shown similar to the manner claim 1 of Ben-David and Litman (1998) was proved), combining this with Lemma 5 completes the proof. $\square$

# 4    Analysis of Realizable PU Learning –Beyond SCAR

In this section, we study PU learning under the realizability assumption when positive examples are sampled from a distribution $\mathcal{P}$ which can differ from $\mathcal{D}_+$. Throughout this section we consider distributions $\mathcal{D}$ with deterministic labels, i.e., $\mathcal{D}(y = 1 \mid x)$ is always zero or one for every $x \in \mathcal{X}$, and we define $l(x) := \mathcal{D}(y = 1 \mid x)$ to be the *labeling function*. We study two classes of distributions for sampling positive examples $\mathcal{P}$:

(i) Selected at random (SAR): For any distribution $e$ over $\mathcal{X}$, define $\mathcal{D}_e(A) = \int \mathcal{D}_+(A) de$, and $\mathcal{P}$ belongs to
$$\mathcal{K}_{\mathcal{D}}^{sar} := \{\mathcal{D}_e \mid \text{any distribution } e \text{ over } \mathcal{X}\}.$$

(ii) Positive covariate shift (PCS): $\mathcal{P}$ belongs to
$$\mathcal{K}_{\mathcal{D}}^{cov} := \{\mathcal{P} \mid \mathcal{P}(A) = 0 \text{ if } \mathcal{D}_+(A) = 0 \text{ and } \mathcal{D}(A) > 0, \ A \text{ is measurable set}\}$$

Note that the condition $\mathcal{P} \in \mathcal{K}_{\mathcal{D}}^{sar}$ is equivalent to having $\mathcal{P}(A) = 0$ when $\mathcal{D}_+(A) = 0$ for any measurable set $A$, i.e., support of $\mathcal{P}$ being a subset of the support of $\mathcal{D}_+$. Thus, $\mathcal{K}_{\mathcal{D}}^{cov}$ is a generalization of the previous case.

We begin by analyzing the simpler case where $\mathcal{P} \in \mathcal{K}_{\mathcal{D}}^{sar}$, and then extend our results to the more general setting $\mathcal{K}_{\mathcal{D}}^{cov}$. Even when $\mathcal{P}$ belongs to $\mathcal{K}_{\mathcal{D}}^{sar}$, additional assumptions on $\mathcal{P}$ are required for the PU learning problem to be well-posed. For example, consider the case where $\mathcal{P}$ is a single point mass on a positively labeled instance. In this scenario, the PU learner would only observe one labeled example, rendering the learning task trivial and unsolvable. To avoid such cases, we impose a common assumption when dealing with distribution shift: a bounded *weight ratio* between $\mathcal{P}$ and $\mathcal{D}_+$. The weight ratio is formally defined as follows.

**Definition 4** (weight ratio). *Let $\mathcal{B} \subseteq 2^X$ be a collection of subsets of the domain $\mathcal{X}$ measurable with respect to both $\mathcal{Q}_{\mathcal{X},1}$ and $\mathcal{Q}_{\mathcal{X},2}$. We define the weight ratio of the source distribution and the target distribution with respect to $\mathcal{B}$ as*

$$R_{\mathcal{B}}(\mathcal{Q}_{\mathcal{X},1}, \mathcal{Q}_{\mathcal{X},2}) = \inf_{\substack{A \in \mathcal{B}(\mathcal{X}) \\ \mathcal{Q}_{\mathcal{X},2}(A) \neq 0}} \frac{\mathcal{Q}_{\mathcal{X},1}(A)}{\mathcal{Q}_{\mathcal{X},2}(A)},$$

*We denote the weight ratio with respect to the collection of all sets that are $\mathcal{Q}_1$ and $\mathcal{Q}_2$-measurable by $R(\mathcal{Q}_1, \mathcal{Q}_2)$.*

Our sample complexity upper bound for the case where $\mathcal{P} \in \mathcal{K}_{\mathcal{D}}^{sar}$ is the direct implication of Lemma 7 proven by Ben-David and Urner (2012), which we state below

**Lemma 7** (Lemma 3 of Ben-David and Urner (2012))**.** *Let $\mathcal{X}$ be some domain, $\mathcal{B} \subseteq 2^{\mathcal{X}}$ a collection of subsets of $\mathcal{X}$, and $\mathcal{Q}_{\mathcal{X},1}$ and $\mathcal{Q}_{\mathcal{X},2}$ distributions over $\mathcal{X}$ with $R := R_{\mathcal{B}}(\mathcal{Q}_{\mathcal{X},1}, \mathcal{Q}_{\mathcal{X},2}) \geq 0$. Then every $R\varepsilon$-net for $\mathcal{B}$ with respect to $\mathcal{Q}_{\mathcal{X},1}$ is an $\varepsilon$-net for $\mathcal{B}$ w.r.t. $\mathcal{Q}_{\mathcal{X},2}$.*

**Theorem 8.** *Let $\mathcal{C}$ be a concept class over domain $\mathcal{X}$ with VC dimension $d$ and $r \in (0, 1)$. Let $\mathcal{W}$ be a set of duos $(\mathcal{P}, \mathcal{D})$ such that $\mathcal{D}$ is realized by $\mathcal{C}$, $\mathcal{P} \in \mathcal{K}_{\mathcal{D}}^{sar}$, and $R_{\mathcal{C}\Delta\mathcal{C}}(\mathcal{P}, \mathcal{D}_+) \geq r$. Then PERM algorithm (1) PU learns $\mathcal{C}$ over $\mathcal{W}$ with sample complexity $m_{\mathcal{C}}^{unlabel}(\varepsilon, \delta) = O\left(\frac{d \ln(1/\varepsilon) + \ln(1/\delta)}{\varepsilon}\right)$ and $m_{\mathcal{C}}^{pos}(\varepsilon, \delta) = O\left(\frac{d \ln(1/r\varepsilon) + \ln(1/\delta)}{r\varepsilon}\right)$.*

Next we derive a nearly tight lower bound for the sample complexity of positive examples when $\mathcal{P} \in \mathcal{K}_{\mathcal{D}}^{sar}$ and $R(\mathcal{P}, \mathcal{D}_+) \geq r$.

**Theorem 9.** *Let $\mathcal{C}$ be a concept class over domain $\mathcal{X}$ with VC dimension $d \geq 2$ and $r \in (0, 1)$. There exists a $M > 1$ such that for any number of positive samples upper bounded by $b \leq M\left(\frac{d + \ln(1/\delta)}{r\varepsilon}\right)$ and any number of unlabeled samples $a \in \mathbb{N}$, $\varepsilon, \delta \in (0, 1) \times (0, 1)$, and PU learner $\mathcal{A}$, there is a distribution $\mathcal{D}$ realized by $\mathcal{C}$ over $\mathcal{X} \times \{0, 1\}$ and a distribution $\mathcal{P} \in \mathcal{K}_{\mathcal{D}}^{sar}$ such that $R(\mathcal{P}, \mathcal{D}_+) \geq r$ and $\Pr_{S^P \sim \mathcal{P}^b, S^U \sim \mathcal{D}_{\mathcal{X}}^a}\left[\text{err}_{\mathcal{D}}(\mathcal{A}(S^P, S^U)) \geq \varepsilon\right] > \delta$.*

The proof of Theorem 9 closely follows that of Theorem 1 (see appendix). Now, we can shift our focus to $\mathcal{P} \in \mathcal{K}_{\mathcal{D}}^{cov}$. In Theorem 10, inspired by Ben-David and Urner (2012) we show that for $\mathcal{P} \in \mathcal{K}_{\mathcal{D}}^{cov}$ no weight ratio assumption is sufficient for PU learnability, unless the total number

of positive and unlabeled samples depends on the size of the domain. Therefore, similar to Ben-David and Urner (2012) in the cases where $\mathcal{P} \in \mathcal{K}_{\mathcal{D}}^{cov}$, in addition to a weight ratio assumption, we assume that the labeling function $l$ is a $\gamma-$margin classifier w.r.t. $\mathcal{D}$, and $\mathcal{D}$ is *realizable by $\mathcal{C}$ with margin $\gamma$*. These notions are formally defined in the following. Moreover, we also assume that $\mathcal{D}[\{(x,1) : x \in \mathcal{X}\}]$ has a constant lower bound (note that we are not assuming $\mathcal{D}[\{(x,1) : x \in \mathcal{X}\}]$ is known).

**Definition 5** (realizable with $\gamma-$margin). *Let $\mathcal{X} \subseteq \mathbb{R}^k, \mathcal{D}$ be a distribution over $\mathcal{X} \times \{0,1\}$ and $c : \mathcal{X} \to \{0,1\}$ a classifier. For all $x \in \mathcal{X}$, denote $B_\gamma(x)$ as the norm-2 ball with radius $\gamma$ centered on $x$. We say that $c$ is a $\gamma$-margin classifier with respect to $\mathcal{D}_{\mathcal{X}}$ if for all $x \in \mathcal{X}$ whenever $\mathcal{D}_{\mathcal{X}}(B_\gamma(x)) > 0$ then $c(y) = c(z)$ holds for all $y, z \in B_\gamma(x)$. We say that a class $\mathcal{C}$ realizes $\mathcal{D}$ with margin $\gamma$ if the optimal (zero-error) classifier $c^\star$ is a $\gamma$-margin classifier.*

Note that a function $c$ being a $\gamma$-margin classifier with respect to $\mathcal{D}_{\mathcal{X}}$ is equivalent to $c$ satisfying the Lipschitz property with Lipschitz constant $1/2\gamma$ on the support of $\mathcal{D}_{\mathcal{X}}$.

**Theorem 10.** *Consider any finite domain $\mathcal{X}$. There exists a concept class $\mathcal{C}_{0,1}$ with $\mathrm{VCD}(\mathcal{C}_{0,1}) = 1$, such that for every PU learner $\mathcal{A}$, and $\varepsilon$ and $\delta$ with $2\varepsilon + \delta < 1/2$, $b, a \in \mathbb{N}$ such that the total number of positive and unlabeled data is upper bounded by $b + a < \sqrt{\frac{2(1-2(2\varepsilon+\delta))|\mathcal{X}|}{3}} - 2$, there exists a distribution $\mathcal{D}$ over $\mathcal{X} \times \{0,1\}$ with deterministic labels which is realized by $\mathcal{C}_{0,1}$ and $\mathcal{P} \in \mathcal{K}_{\mathcal{D}}^{cov}$ where $R(\mathcal{P}, \mathcal{D}_+) = 1/2$, $\mathcal{D}[\{(x,1) : x \in \mathcal{X}\}] \geq 1/2$ and $\mathrm{Pr}_{S^P \sim \mathcal{P}^b, S^U \sim \mathcal{D}_{\mathcal{X}}^a}\left[\mathrm{err}_{\mathcal{D}}(\mathcal{A}(S^P, S^U) \geq \varepsilon\right] > \delta$.*

The following theorem is inspired by Theorem 2 of Ben-David and Urner (2012), which establishes a lower bound on sample size for infinite domains under the additional assumptions that the labeling function is $\lambda$-Lipschitz and that $\mathcal{D}[\{(x,1) : x \in \mathcal{X}\}] \geq \frac{1}{2}$. As shown, even with these additional assumptions, the total number of samples must be at least exponential in the Lipschitz constant. This can be viewed as the additional cost incurred when $\mathcal{P} \in \mathcal{K}_{\mathcal{D}}^{cov}$.

**Theorem 11.** *Let $\mathcal{X} = [0,1]^k$. There exists a concept class $\mathcal{C}_{0,1}$ with $\mathrm{VCD}(\mathcal{C}_{0,1}) = 1$, such that for every PU learner $\mathcal{A}$, and $\varepsilon$ and $\delta$ with $2\varepsilon + \delta < 1/2$, $b, a \in \mathbb{N}$ such that the total number of positive and unlabeled data is upper bounded by $b + a < \sqrt{\frac{2(1+\lambda)^k(1-2(2\varepsilon+\delta))}{3}} - 2$, there exists a distribution $\mathcal{D}$ over $\mathcal{X} \times \{0,1\}$ with deterministic labels which is realized by $\mathcal{C}_{0,1}$ and $\mathcal{P} \in \mathcal{K}_{\mathcal{D}}^{cov}$ where $C(\mathcal{P}, \mathcal{D}_+) = 1/2$, $\mathcal{D}[\{(x,1) : x \in \mathcal{X}\}] \geq 1/2$ and $l$ is a $\lambda$-Lipschitz labeling function and $\mathrm{Pr}_{S^P \sim \mathcal{P}^b, S^U \sim \mathcal{D}_{\mathcal{X}}^a}\left[\mathrm{err}_{\mathcal{D}}(\mathcal{A}(S^P, S^U) \geq \varepsilon\right] > \delta$.*

Next, we present Algorithm 1, designed for the case where $\mathcal{P} \in \mathcal{K}_{\mathcal{D}}^{cov}$. The algorithm is inspired by the domain adaptation method introduced in Ben-David and Urner (2012). In the standard domain adaptation setting, the goal is to minimize the error with respect to a target distribution $\mathcal{Q}_T$, given labeled samples from a source distribution $\mathcal{Q}_S$ and unlabeled samples from $\mathcal{Q}_T$.

Algorithm 1 adapts this approach to the PU learning setting, with $\mathcal{Q}_S = \mathcal{P}$ and $\mathcal{Q}_T = \mathcal{D}_+$ (with labels being 1). However, unlike domain adaptation, PU learning lacks access to unlabeled samples from $\mathcal{D}_+$; instead, it only has access to unlabeled samples from $\mathcal{D}_{\mathcal{X}}$. To account for this difference, two key modifications are made to the algorithm from Ben-David and Urner (2012): (i) Instead of using a sample $T$ from $\mathcal{D}_+$, Algorithm 1 uses the unlabeled sample $S^U$ drawn from $\mathcal{D}_{\mathcal{X}}$; (ii) The algorithm outputs a PERM rather than an ERM.

Notice that in Theorem 12, we require the number of unlabeled samples to be exponential with respect to $1/\gamma$ (as it was required for the total number of samples to be exponential with respect to the Lipschitz constant in our lower bound appearing in Theorem 11). However, in many learning scenarios, unlabeled data is abundantly available while labeled data is difficult to obtain, which makes this algorithm more practically appealing.

**Theorem 12.** *Let $\mathcal{X} = [0,1]^k, \gamma > 0$ a margin parameter, $\pi, r > 0$ and $\mathcal{C}$ be a realizable concept class with VC dimension $d < \infty$. Let $\mathcal{W}$ to be the set of duos $(\mathcal{P}, \mathcal{D})$ such that:*

- *$\mathcal{P} \in \mathcal{K}_{\mathcal{D}}^{cov}$, $\mathcal{D}$ is realizable by $\mathcal{C}$ with margin $\gamma$ and has deterministic labels, and $\mathcal{D}(y = 1) \geq \pi$.*

- *The labeling function $l$ is a $\gamma$-margin classifier with respect to $\mathcal{D}_{\mathcal{X}}$.*

- *$R_{\mathcal{I}}(\mathcal{P}, \mathcal{D}_+) \geq r$ for the class $\mathcal{I} = (\mathcal{C}\Delta\mathcal{C}) \sqcap \mathcal{B}$, where $\mathcal{B}$ is a partition of $[0,1]^k$ into boxes of sidelength $\gamma/\sqrt{k}$.*

**Algorithm 1:** Algorithm for PU learning in the positive covariate shift setup

---

**Input:** $S^P$ i.i.d. sampled from $\mathcal{P}$ with label 1 and an unlabeled i.i.d. sample $S^U$ from $\mathcal{D}_{\mathcal{X}}$ and a margin parameter $\gamma$.

1 Partition the domain $[0,1]^k$ into a collection $\mathcal{B}$ of boxes (axis-aligned rectangles) with sidelength $(\gamma/\sqrt{k})$ ;

2 Obtain sample $S'$ by removing every point in $S^P$, which is sitting in a box that is not hit by $S^U$ ;

3 **return** $\operatorname{argmin}_{c \in \mathcal{C}, Domain(S') \subseteq c} |c \mid S^U|$

---

*Then Algorithm 1 PU learns $\mathcal{C}$ over $\mathcal{W}$ with sample complexity*

$$m_{\mathcal{C}}^{pos}(\varepsilon, \delta) = O\left(\frac{d \ln\left(1/(r(1-\varepsilon)\varepsilon)\right) + \ln(1/\delta)}{r^2(1-\varepsilon)^2 \varepsilon}\right),$$

$$m_{\mathcal{C}}^{unlabel}(\varepsilon, \delta) = O\left(\frac{(\sqrt{k}/\gamma)^k \ln\left((\sqrt{k}/\gamma)^k/\delta\right)}{\pi \varepsilon} + \frac{d\ln(1/\varepsilon) + \ln(1/\delta)}{\varepsilon}\right).$$

## 5 Analysis of the Agnostic PU Learning

In this section we analyze agnostic PU learning. It is already known that with the knowledge of class prior $\alpha$, every class with finite VC dimension is PU learnable Du Plessis et al. (2015) in the SCAR setup. In Section 5.1 we derive a nearly matching lower bound on both the sample complexity of unlabeled examples and positive examples to Du Plessis et al. (2015) upper bounds. Then, we show that for a concept class $\mathcal{C}$ with more than two concepts, without the knowledge of $\alpha$, no PU learner–without access to $\alpha$–can achieve an error less than $\frac{\max(\alpha, 1-\alpha)}{\min(\alpha, 1-\alpha)}$ times the approximation error even in the SCAR setup (which makes the PU learning task impossible). Furthermore, in Section 5.2, we complement this result by showing that for every concept class, there exists an algorithm whose error is arbitrarily close to $\frac{\max(\alpha, 1-\alpha)}{\min(\alpha, 1-\alpha)}$ times the approximation error in the SCAR setup. Finally, we derive generalization bounds for settings where $S^P$ is drawn from an arbitrary distribution $\mathcal{P}$.

### 5.1 Lower Bounds –SCAR setup

The following theorem provides an almost tight lower bound on the sample complexity of both positive and unlabeled examples, assuming the learner knows that $\alpha = \frac{1}{2}$. We prove this theorem by reducing it to a problem called *the generalized weighted die problem*, which is a problem inspired by Ben-David and Ben-David (2011). Detailed Proof of the theorem is deferred to the appendix.

**Theorem 13.** *Let $\mathcal{C}$ be a concept class with $\mathrm{VCD}(\mathcal{C}) = d$ where $d \geq 4$. Consider $\mathcal{W}$ to be the set of duos $(\mathcal{D}, \mathcal{D}_+)$ with $\mathcal{D}[\{(x,1) : x \in \mathcal{X}\}] = 0.5$. Then $\mathcal{C}$ is PU learnable over $\mathcal{W}$ with sample complexity $m_{\mathcal{C}}^{unlabel}(\varepsilon, \delta), m_{\mathcal{C}}^{pos}(\varepsilon, \delta) = \Omega\left(\frac{d + \ln(1/\delta)}{\varepsilon^2}\right)$ and $m_{\mathcal{C}}^{unlabel}(\varepsilon, \delta), m_{\mathcal{C}}^{pos}(\varepsilon, \delta) = O\left(\frac{d \ln(1/\varepsilon) + \ln(1/\delta)}{\varepsilon^2}\right)$.*

Next, we present a lower bound on the generalization of PU learners for the cases where $\alpha$ is unknown.

**Theorem 14.** *Let $\mathcal{C}$ be a concept class over $\mathcal{X}$ containing at least two distinct concepts. Then, for every $\eta \in (0,1)$, any number of positive samples $b \in \mathbb{N}$, any number of unlabeled samples $a \in \mathbb{N}$, and PU learner $\mathcal{A}$, there exists a distribution $\mathcal{D}$ over $\mathcal{X} \times \{0,1\}$ with $\alpha \in \{\eta, 1-\eta\}$, where $\alpha := \mathcal{D}[\{(x,1) : x \in \mathcal{X}\}]$, such that*

$$\Pr_{S^P \sim \mathcal{D}_+^b, S^U \sim \mathcal{D}_{\mathcal{X}}^a}\left[\mathrm{err}_{\mathcal{D}}\left(\mathcal{A}(S^P, S^U)\right) \geq \frac{\max(\alpha, 1-\alpha)}{\min(\alpha, 1-\alpha)} \min_{c \in \mathcal{C}} \mathrm{err}_{\mathcal{D}}(c)\right] = 1.$$

*Proof.* Let $x$ be any instance such that two concepts in $\mathcal{C}$ disagree on its label. Define distribution $\mathcal{D}_0$ over $\mathcal{X} \times \{0,1\}$ to assign probability $\eta$ on $(x,1)$ and $1 - \eta$ over $(x,0)$, and $\mathcal{D}_1 := 1 - \mathcal{D}_0$. Then

for any $z \in \{0, 1\}$ we have $\min_{c \in \mathcal{C}} \mathrm{err}_{\mathcal{D}_z}(c) = \min(\eta, 1 - \eta)$ and $\mathcal{D}_{+,z} = \mathcal{D}_{\mathcal{X},z} = \mathbb{1}_{\{x\}}$. Thus, for any $b, a \in \mathbb{N}$ and $S^P \sim \mathcal{D}^b_{+,z}, S^U \sim \mathcal{D}^a_{\mathcal{X},z}$ there exists a $z \in \{0, 1\}$ such that $\mathrm{err}_{\mathcal{D}_z}\left(\mathcal{A}(S^P, S^U)\right) = \max(\eta, 1 - \eta)$. This completes the proof. $\qquad\square$

**Remark 15.** *Our poof also demonstrates that, even when the approximation error is known, almost no concept class is PU learnable. This finding is particularly noteworthy because agnostic PU learning with known approximation error can be viewed as a relaxation of the realizable PU learning setting.*

## 5.2 Upper bounds

We start by proposing an algorithm for agnostic PU learning that, for a given $\gamma > 0$, outputs a concept which minimizes the *Lagrangian PU empirical loss* $\hat{\mathrm{err}}^\gamma : \mathcal{C} \to \mathbb{R}^{\geq 0}$, defined as

$$\hat{\mathrm{err}}^\gamma(c) := \frac{\left|c \mid S^U\right|}{|S^U|} + \gamma \cdot \frac{|S^P| - \left|c \mid S^P\right|}{|S^P|}. \tag{2}$$

Note that the PERM algorithm minimizes $\frac{|c|S^U|}{|S^U|}$ while assuming that the empirical error of $c$ (for realizable concept classes) is zero on $S^P$. Since $\frac{|S^P| - |c|S^P|}{|S^P|}$ is the empirical error on $S^P$, this algorithm can also be viewed as a Lagrangian function for the PERM algorithm. Also, notice that when $\gamma = 2\alpha$, $\hat{\mathrm{err}}^\gamma$ will be equivalent to the surrogate loss introduced in Du Plessis et al. (2015) when the loss function is the zero-one loss. We begin by analyzing the SCAR setup.

**Theorem 16.** *Let $\mathcal{C}$ be any concept class over domain $\mathcal{X}$ with VC dimension $d$, and let $\mathcal{P} = \mathcal{D}_+$. Given any $\gamma \geq \alpha$, denote $c^{PU} = \mathrm{argmin}_{c \in \mathcal{C}} \hat{\mathrm{err}}^\gamma(c)$. There exists $M > 1$ such that for all $c \in \mathcal{C}$, if $|S^P|, |S^U| > \frac{M(d + \ln(1/\delta))}{\varepsilon^2}$, then with probability $1 - 4\delta$ we have*

$$\mathrm{err}_{\mathcal{D}}(c^{PU}) \leq \max\left(\frac{\gamma - \alpha}{\alpha}, \frac{\alpha}{\gamma - \alpha}\right)(\mathrm{err}_{\mathcal{D}}(c) + 2(1 + \gamma)\varepsilon)$$

**Remark 17.** *Let's also suppose as a prior knowledge we have access to $\hat{\alpha} \approx \alpha$ where $2\hat{\alpha} \geq \alpha$. Then one can incorporate the prior knowledge by setting $\gamma = 2\hat{\alpha}$. In particular, if $\alpha$ was known with $\gamma = 2\alpha$, we would have $\mathrm{err}_{\mathcal{D}}(c^{PU}) \leq \mathrm{err}_{\mathcal{D}}(c) + 6\varepsilon$. This is consistent with Du Plessis et al. (2015) results for cases where the class prior is known.*

The following corollary is a direct consequence of applying Theorem 16 with $\gamma = 1$.

**Corollary 18.** *For any concept class $\mathcal{C}$ with VC dimension $d$, there exists a PU learner $\mathcal{A}$ and a constant $M > 1$ such that for every $\alpha, \varepsilon, \delta \in (0, 1)$ and for all $b, a > \frac{M(d + \ln(1/\delta))}{\varepsilon^2}$, and for any distribution $\mathcal{D}$ over $\mathcal{X} \times \{0, 1\}$ with $\mathcal{D}[\{(x, 1) : x \in \mathcal{X}\}] = \alpha$, the following holds: for any sample $S^P$ of size $b$ drawn i.i.d. from $\mathcal{D}_+$ and any sample $S^U$ of size $a$ drawn i.i.d. from $\mathcal{D}_{\mathcal{X}}$, with probability at least $1 - \delta$,*

$$\mathrm{err}_{\mathcal{D}}\left(\mathcal{A}(S^P, S^U)\right) \leq \frac{\max(\alpha, 1 - \alpha)}{\min(\alpha, 1 - \alpha)}\left(\min_{c \in \mathcal{C}} \mathrm{err}_{\mathcal{D}}(c) + 4\varepsilon\right).$$

Finally, we examine the most general PU learning setting. We derive generalization bounds that hold for arbitrary concept classes and any distribution $\mathcal{P}$ by combining Theorem 16 with Ben-David et al. (2010) results. These bounds involve the $\mathcal{C}\Delta\mathcal{C}$ distance, which we formally define below.

**Definition 6.** *Kifer et al. (2004) Given a domain $\mathcal{X}$ and a collection $\mathcal{B}$ of subsets of $\mathcal{X}$, let $\mathcal{Q}_{\mathcal{X},1}, \mathcal{Q}_{\mathcal{X},2}$ be probability distributions over $\mathcal{X}$, such that every set in $\mathcal{B}$ is measurable with respect to both distributions. The $\mathcal{B}$-distance between such distributions is defined as*

$$d_{\mathcal{B}}\left(\mathcal{Q}_{\mathcal{X},1}, \mathcal{Q}_{\mathcal{X},2}\right) = 2 \sup_{B \in \mathcal{B}} \left|\Pr_{\mathcal{Q}_{\mathcal{X},1}}[B] - \Pr_{\mathcal{Q}_{\mathcal{X},2}}[B]\right|$$

**Theorem 19.** *Let $\mathcal{C}$ be any concept class over domain $\mathcal{X}$ with VC dimension $d$, and let $\mathcal{P}$ be any arbitrary distribution. Given any $\gamma \geq \alpha$, denote $c^{PU} = \mathrm{argmin}_{c \in \mathcal{C}} \hat{\mathrm{err}}^\gamma(c)$. There exists $M > 1$ such that for all $c \in \mathcal{C}$, if $|S^P|, |S^U| > \frac{M(d + \ln(1/\delta))}{\varepsilon^2}$, then with probability $1 - 4\delta$ we have*

$$\mathrm{err}_{\mathcal{D}}(c^{PU}) \leq \max\left(\frac{\gamma - \alpha}{\alpha}, \frac{\alpha}{\gamma - \alpha}\right)\left(err_{\mathcal{D}}(c) + 2(1 + \gamma)\varepsilon + 2\gamma\left(\lambda^P + d_{\mathcal{C}\triangle\mathcal{C}}(\mathcal{P}, \mathcal{D}_+)\right)\right)$$

*Where $\lambda^P := \min_{c \in \mathcal{C}}\left(\mathrm{err}^+_{\mathcal{D}}(c) + \mathrm{err}_{\mathcal{P}}(c, 1)\right)$ and $\mathrm{err}_{\mathcal{P}}(c, 1) := \Pr_{x \sim \mathcal{P}}(c(x) \neq 1)$.*

# 6 Conclusion

In conclusion, this work studies the sample complexity of PU learning in both realizable and agnostic settings, covering the SCAR setup as well as more general scenarios. We provide theoretical guarantees on finite sample complexity. Our results extend the existing literature by relaxing several restrictive assumptions that were made in previous publications, and by proving lower bounds on required sample sizes.

## Acknowledgments and Disclosure of Funding

We thank Sandra Zilles and Alireza Fathollah Pour for helpful discussions during the development of this work, and the anonymous reviewer for pointing out the notions of slicing dimension and 1-centered star number.

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

# A  Useful Theorems

**Lemma 20.** *Let $Z$ be a random variable such that $Z \in [0, \gamma]$ and $\Pr[Z > \varepsilon] \leq \delta$. Then $\mathbb{E}[Z] < \gamma\delta + \varepsilon(1-\delta)$.*

**Lemma 21** (Multiplicative Chernoff bounds Motwani and Raghavan (1996))**.** *Let $X_1, \ldots, X_m$ be independent random variables drawn according to some distribution $\mathcal{D}$ with mean $p$ and support included in $[0, 1]$. Then, for any $\gamma \in \left[0, \frac{1}{p} - 1\right]$, the following inequality holds for $\widehat{p} = \frac{1}{m}\sum_{i=1}^{m} X_i$:*

$$\mathbb{P}[\widehat{p} \geq (1 + \gamma)p] \leq e^{-\frac{mp\gamma^2}{3}}$$

$$\mathbb{P}[\widehat{p} \leq (1 - \gamma)p] \leq e^{-\frac{mp\gamma^2}{2}}$$

**Theorem 22** (Hoeffding Inequality Hoeffding (1994)). *Let $X_1, \ldots, X_n$ be independent random variables such that $a_i \leq X_i \leq b_i$ almost surely. Consider the sum of variables,*

$$S_n = X_1 + \cdots + X_n$$

*Then Hoeffding's theorem states that, for all $t > 0$,*

$$\Pr\left(|S_n - \mathrm{E}\left[S_n\right]| \geq t\right) \leq 2 \exp\left(-\frac{2t^2}{\sum_{i=1}^{n}(b_i - a_i)^2}\right)$$

**Theorem 23** (Chebyshev's inequality Feller (1991)). *Let $X$ be a random variable with bounded non-zero variance. Then for any $k > 0$,*

$$\Pr(|X - \mathbb{E}\left[X\right]| \geq k) \leq \frac{\mathrm{Var}[X]}{k^2}$$

**Lemma 24** (Slud's inequality Slud (1977)). *For $S \sim \mathrm{Bin}(m, p)$ where $p \leq \frac{1}{2}$ and $b$ is an integer with $mp \leq b \leq m(1 - p)$ then*

$$P[S \geq b] \geq P\left[Z \geq \frac{b - mp}{\sqrt{mp(1 - p)}}\right]$$

*where $Z \sim N(0, 1)$ is a normally distributed random variable with mean of 0 and standard deviation of 1.*

**Lemma 25** (Normal tail bound Tate (1953)). *For standard Gaussian random variable $Z \sim N(0, 1)$ and $x \geq 0$ we have*

$$P[Z \geq x] \geq \frac{1}{2}\left(1 - \sqrt{1 - e^{-x^2}}\right)$$

## B   Missing Proofs from Section 3

**Theorem 1.** *Let $\mathcal{C}$ be a concept class with VC dimension $d \geq 2$ over the domain $\mathcal{X}$. There exists a $M > 1$ such that for any number of positive samples upper bounded by $b \leq M\left(\frac{d + \ln(1/\delta)}{\varepsilon}\right)$ and for every number of unlabeled samples $a \in \mathbb{N}$, $(\varepsilon, \delta) \in (0, 1) \times (0, 1)$, and PU learner $\mathcal{A}$ there is a distribution $\mathcal{D}$ realized by $\mathcal{C}$ over $\mathcal{X} \times \{0, 1\}$ such that $\Pr_{S^P \sim \mathcal{D}_+^b, S^U \sim \mathcal{D}_{\mathcal{X}}^a}\left[\mathrm{err}_{\mathcal{D}}(\mathcal{A}(S^P, S^U)) \geq \varepsilon\right] > \delta.$*

*Proof.* Proof of $m_{\mathcal{C}}^{pos}(\varepsilon, \delta) = \Omega(\frac{d}{\varepsilon})$. We prove that for $d \geq 9$ we have $m_{\mathcal{C}}^{pos}(\varepsilon, \frac{1}{2000}) \geq \frac{d-1}{32000\varepsilon}$. Let $B = \{x_1, ..., x_d\}$ be a set of size $d$ shattered by $\mathcal{C}$, and $\varepsilon = \min\left(\frac{d-1}{32000b}, 0.0005\right)$ and $\rho = 2000\varepsilon$. Denote $\bar{B} := B \setminus \{x_d\}$, and for any $O \subseteq \bar{B}$ define $\mathcal{D}_O$ over $X \times \{0, 1\}$ be

$$\mathcal{D}_O(\{(x, y)\}) := \begin{cases} 1 - \rho & x = x_d \text{,and } y = 1 \\ \frac{\rho}{d-1} & x \in O, \text{ and } y = 1 \\ \frac{\rho}{d-1} & x \notin O, \text{ and } y = 0 \\ 0 & \text{o.w.} \end{cases} \tag{3}$$

Define $\mathcal{W}_{\rho, d}^{scar-pos} := \{(\mathcal{D}_O, \mathcal{D}_{+,O}) \mid O \subseteq \bar{B}\}$. Note that for proving the claim it is enough to show that for every PU learner $\mathcal{A}$, there exists a $O^\star \subseteq \bar{B}$ such that

$$\Pr_{S^P \sim \mathcal{D}_{+,O^\star}^b, S^U \sim \mathcal{D}_{\mathcal{X},O^\star}^a}\left[\mathrm{err}_{\mathcal{D}_{O^\star}}\left(\mathcal{A}(S^P, S^U)\right) \geq \varepsilon\right] > \frac{1}{500}.$$

Note that, for all $O, O' \subseteq B$ we have $\mathcal{D}_{\mathcal{X},O} = \mathcal{D}_{\mathcal{X},O'}$. Therefore, for this set of distributions, the unlabeled sample does not help the learner, and it doesn't affect the proof. Thus, for the sake of simplicity, we shorten $\mathcal{A}(S^P, S^U)$ to $\mathcal{A}(S^P)$. Also, without loss of generality, we can assume that $\mathcal{A}$ always predicts 1 on instance $x_d$.

From this point on for any sample $S$ with $Domain(S) = B$, denote $\bar{S} := S \setminus \{x_d\}$. Next, define event $E$ to be the event that $|O| \geq \frac{d-1}{4}$ and $|\bar{S}^P| \leq \frac{d-1}{8}$. Since maximum is no less than the average, we have

$$
\max_{O \subseteq \bar{B}} \mathbb{E}_{S^P \sim \mathcal{D}^b_{+,O}} \left[ \mathrm{err}_{\mathcal{D}_O} \left( \mathcal{A}(S^P) \right) \right]
$$

$$
\geq \mathbb{E}_{O \sim U_{2^{\bar{B}}}, S^P \sim \mathcal{D}^b_{+,O}} \left[ \mathrm{err}_{\mathcal{D}_O} \left( \mathcal{A}(S^P) \right) \right] \tag{4}
$$

$$
\geq \mathbb{E}_{O \sim U_{2^{\bar{B}}}, S^P \sim \mathcal{D}^b_{+,O}} \left[ \mathrm{err}_{\mathcal{D}_O} \left( \mathcal{A}(S^P) \right) \mid E \right] \Pr_{O \sim U_{2^{\bar{B}}}, S^P \sim \mathcal{D}^b_{+,O}} [E]
$$

We first derive a lower bound for $\Pr_{O \sim U_{2^{\bar{B}}}, S^P \sim \mathcal{D}^b_{+,O}}[E]$. First, note that since $O \sim U_{2^{\bar{B}}}$ we have $|O| \sim Bin(d-1, \frac{1}{2})$. Thus, using the Multiplicative Chernoff bound, as long as $d \geq 9$

$$
Pr_{O \sim U_{2^{\bar{B}}}} \left[ |O| < \frac{d-1}{4} \right] \geq \left( 1 - \exp\left( -\frac{d-1}{16} \right) \right) > 0.3 \tag{5}
$$

Next fix any $O \subseteq \bar{B}$ with $|O| \geq \frac{d-1}{4}$. For every $i$ such that $x_i \in O$, and $j \in [b]$ let the random variable $Y_{i,j}$ be 1 if the $j$th sample drawn from $\mathcal{D}_{+,O}$ is $x_i$ and 0 otherwise. Note that $Y_{i,j}$ is simply a Bernoulli with parameter at least $\frac{\rho}{d-1}$. Then, $|\bar{S}^P| = \sum_{i:x_i \in O} \sum_{j=1}^{b} Y_{i,j}$. Therefore, using the Multiplicative Chernoff bound (Lemma 21) for any $\gamma \in \left[ 0, \frac{(d-1)}{|O|} - 1 \right]$ we have

$$
\Pr_{S \sim \tilde{\mathcal{D}}^b_{+,O}} \left[ |\bar{S}^P| \geq (1+\gamma)b\rho \right] \leq e^{-\frac{b\rho\gamma^2|O|}{3(d-1)}} \leq e^{-\frac{b\rho\gamma^2}{12}}
$$

Set $\gamma = 1$. Note that $\rho \leq \frac{d-1}{16b}$. Thus

$$
\Pr_{S \sim \tilde{\mathcal{D}}^b_{+,O}} \left[ |\bar{S}^P| > \frac{d-1}{8} \right] \leq e^{-\frac{d-1}{192}} < 0.96 \tag{6}
$$

Combining (5) and (6) we derive that $\Pr_{O \sim U_{2^{\bar{B}}}, S^P \sim \mathcal{D}^b_{+,O}}[E] > 0.012$.

Next we try to derive a lower bound for $\mathbb{E}_{O \sim U_{2^{\bar{B}}}, S^P \sim \mathcal{D}^b_{+,O}} \left[ \mathrm{err}_{\mathcal{D}_O} \left( \mathcal{A}(S^P) \right) \mid E \right]$. Note that for a given $S^P$, due to symmetry for all $O, O' \subseteq \bar{B}$ such that $Domain(\bar{S}^P) \subseteq O, O'$ and $|O| = |O'|$ we have

$$
\Pr_{S \sim \mathcal{D}^b_{+,O}} [S = S^P] = \Pr_{S \sim \mathcal{D}^b_{+,O'}} [S = S^P]
$$

Moreover, it is clear that for $O, O' \subseteq \bar{B}$ such that $Domain(\bar{S}^P) \subseteq O, O'$ and $|O| \geq |O'|$ we also have

$$
\Pr_{S \sim \mathcal{D}^b_{+,O}} [S = S^P] \leq \Pr_{S \sim \mathcal{D}^b_{+,O'}} [S = S^P]
$$

Next fix any $S^P$ with $|\bar{S}^P| \leq \frac{d-1}{8}$. Since given fix $S^P$, smaller $O$ are more likely, due to symmetry we can conclude that given event $E$, for every $x \in \bar{B} \setminus S^P$ the probability of $x \in O$ is at most $\frac{1}{2}$. Moreover, note that for all $|O| \geq \frac{d-1}{4}$ we have

$$
\frac{|O \setminus S^P|}{|\bar{B} \setminus S^P|} \geq \frac{\frac{d-1}{8}}{d-1} = \frac{1}{8}.
$$

Thus, again due to symmetry, given event $E$ for every $x \in \bar{B} \setminus S^P$ the probability of $x \in O$ is at least $\frac{1}{8}$. Thus, no matter what is $\mathcal{A}$ we have

$$
\mathbb{E}_{O \sim U_{2^{\bar{B}}}, S^P \sim \mathcal{D}^b_{+,O}} \left[ \mathrm{err}_{\mathcal{D}_O} \left( \mathcal{A}(S^P) \right) \mid E \right] \geq \frac{|\bar{B} \setminus S^P|\rho}{8(d-1)} \geq \frac{7}{64}\rho
$$

Combining this with (4), we conclude that there exists a $O^\star \subseteq \bar{B}$

$$\mathbb{E}_{S^P \sim \mathcal{D}^b_{+,O^\star}}\left[\mathrm{err}_{\mathcal{D}_{O^\star}}\left(\mathcal{A}(S^P)\right)\right] > \frac{7 * 0.012}{64}\rho > 0.001\rho$$

Note that since $\mathcal{A}$ always predicts the label of $x_d$ to be 1, its error is always less than $\rho$. Therefore, using Lemma 20 we derive

$$\Pr_{S^P \sim \mathcal{D}^b_{+,O^\star}}\left[\mathrm{err}_{\mathcal{D}_{O^\star}}\left(\mathcal{A}(S^P)\right) \geq \varepsilon\right] > \frac{0.001\rho - \varepsilon}{(\rho - \varepsilon)}$$

$$= \frac{\varepsilon}{2000\varepsilon - \varepsilon}$$

$$> \frac{1}{2000}$$

*Proof of* $m^{pos}_\mathcal{C}(\varepsilon, \delta) \geq \Omega\left(\frac{\ln(1/\delta)}{\varepsilon}\right)$. Next, we prove that for every $a \in \mathbb{N}$, $\varepsilon < \frac{1}{2}$, $\delta \in (0,1)$, $b = \frac{\ln\left(\frac{1}{2\delta}\right)}{2\varepsilon}$ and PU learner $\mathcal{A}$ there is a distribution $\mathcal{D}$ realized by $\mathcal{C}$ over $\mathcal{X} \times \{0,1\}$ such that

$$\Pr_{S^P \sim \mathcal{D}^b_+, S^U \sim \mathcal{D}^a_\mathcal{X}}\left[\mathrm{err}_\mathcal{D}\left(\mathcal{A}(S^P, S^U)\right) \geq \varepsilon\right] > \delta.$$

Let $B = \{x_1, x_2\} \subseteq \mathcal{X}$ be any set shattered by $\mathcal{C}$. For $z \in \{0,1\}$ and $(x,y) \in \mathcal{X} \times \{0,1\}$ define $\mathcal{D}_z$ over $\mathcal{X} \times \{0,1\}$ as

$$\mathcal{D}_z(\{(x,y)\}) = \begin{cases} \varepsilon & x = x_1 \text{ and } y = z \\ 1 - \varepsilon & x = x_2 \text{ and } y = 1 \\ 0 & \text{o.w.} \end{cases}$$

Note that both $\mathcal{D}_0$ and $\mathcal{D}_1$ are realized by $\mathcal{C}$. First we try to derive a lower bound for the probability that the sample $S^P \sim \mathcal{D}^b_{+,1}$ doesn't contain $x_1$. Note that, it is easy to see that the function $f(a) = \frac{-\ln(1-a)}{a}$ always has a positive derivative for all $a < 1$. Therefore, for all $\varepsilon < \frac{1}{2}$ we have

$$\frac{-\ln(1-\varepsilon)}{\varepsilon} < 2\ln(2) < 2$$

Thus,

$$\Pr_{S^P \sim \mathcal{D}^b_{+,1}}[x_1 \notin S^P] = (1-\varepsilon)^b = e^{-\varepsilon \frac{-\ln(1-\varepsilon)}{\varepsilon} b} > e^{-2\varepsilon b} = 2\delta. \tag{7}$$

Thus since $\Pr_{S^P \sim \mathcal{D}^b_{+,0}}[x_1 \notin S^P] = 1$. We derive

$$\min_{z \in \{0,1\}} \Pr_{S^P \sim \mathcal{D}^b_{+,z}, S^U \sim \mathcal{D}^a_{\mathcal{X},z}}\left[\mathrm{err}_{D_z}(\mathcal{A}(S^P, S^U)) \geq \varepsilon\right]$$

$$\overset{(i)}{\geq} \min_{z \in \{0,1\}} \Pr_{S^P \sim \mathcal{D}^b_{+,z}, S^U \sim \mathcal{D}^a_{\mathcal{X},z}}\left[\mathcal{A}(S^P, S^U)(x_1) \neq z\right]$$

$$\geq \min_{z \in \{0,1\}} \Pr_{S^P \sim \mathcal{D}^b_{+,z}}\left[S^P = \{x_2\}^b\right] \Pr_{S^U \sim \mathcal{D}^a_{\mathcal{X},z}}\left[\mathcal{A}(\{x_2\}^b, S^U)(x_1) \neq z\right]$$

$$> 2\delta \min_{z \in \{0,1\}} \Pr_{S^U \sim \mathcal{D}^a_{\mathcal{X},z}}\left[\mathcal{A}(\{x_2\}^b, S^U)(x_1) \neq z\right] \tag{8}$$

$$\overset{(ii)}{=} 2\delta \min\left(\Pr_{S^U \sim \mathcal{D}^a_{\mathcal{X},0}}\left[\mathcal{A}(\{x_2\}^b, S^U)(x_1) = 0\right], 1 - \Pr_{S^U \sim \mathcal{D}^a_{\mathcal{X},0}}\left[\mathcal{A}(\{x_2\}^b, S^U)(x_1) = 0\right]\right)$$

$$\geq \delta$$

Where (i) is due to the fact that whenever the learner makes a mistake at $x_1$ the error will be at least $\varepsilon$, and (ii) is due to the fact that $\mathcal{D}_{\mathcal{X},0} = \mathcal{D}_{\mathcal{X},1}$. $\square$

**Theorem 3.** *Let $\mathcal{C}$ be a concept class with claw number $\mathfrak{h} \geq 1$. There exists a $M > 1$ such that for any number of unlabeled samples upper bounded by $a \leq M\left(\frac{\mathfrak{h} + \ln(1/\delta)}{\varepsilon}\right)$ and any number of positive samples $b \in \mathbb{N}$, $(\varepsilon, \delta) \in (0,1) \times (0,1)$, and PU learner $\mathcal{A}$ there is a distribution $\mathcal{D}$ realized by $\mathcal{C}$ over $\mathcal{X} \times \{0,1\}$ such that $\Pr_{S^P \sim \mathcal{D}^b_+, S^U \sim \mathcal{D}^a_\mathcal{X}}\left[\mathrm{err}_\mathcal{D}(\mathcal{A}(S^P, S^U) \geq \varepsilon\right] > \delta.$*

*Proof.* Proof of $m_{\mathcal{C}}^{unlabel}(\varepsilon, \delta) = \Omega\left(\frac{\mathfrak{h}}{\varepsilon}\right)$. First we prove that for every $b, a \in \mathbb{N}$, and PU learner $\mathcal{A}$ there is a distribution $\mathcal{D}$ realized by $\mathcal{C}$ over $\mathcal{X} \times \{0,1\}$ such that

$$\Pr_{S^P \sim \mathcal{D}_{+}^b, S^U \sim \mathcal{D}_{\mathcal{X}}^a} \left[ \text{err}_{\mathcal{D}} \left( \mathcal{A}(S^P, S^U) \right) \geq \min\left[ \frac{7}{2400}, \frac{7\mathfrak{h}}{4800a} \right] \right] > \frac{1}{1000}.$$

Let $\varepsilon := \min\left[ \frac{7}{2400}, \frac{7\mathfrak{h}}{4800a} \right]$, $\rho := \frac{1200\varepsilon}{7}$, and $m := \max\left( 2(b+a), \mathfrak{h}b, \frac{\mathfrak{h}}{\rho} \right) + \mathfrak{h}$. Let $B$ be any set such that $\{ O \subseteq B \mid |O| = m - \mathfrak{h} \} \subseteq \mathcal{C}$. For any $O \subseteq B$ with $|O| = \mathfrak{h}$ and any $(x, y) \in \mathcal{X} \times \{0,1\}$ we define the distribution $\mathcal{D}_O$ over $\mathcal{X} \times \{0,1\}$ as

$$\mathcal{D}_O(\{(x,y)\}) := \begin{cases} \rho/\mathfrak{h} & \text{if } x \in O, \text{ and } y = 0 \\ (1-\rho)/(m-\mathfrak{h}) & \text{if } x \in B \setminus O, \text{ and } y = 1 \\ 0 & \text{o.w.} \end{cases} \quad (9)$$

Note that all such $\mathcal{D}_O$ are realized by $\mathcal{C}$. We prove our claim holds for one of $\mathcal{D}_O$, where $O \in V$. Next, suppose a learner predicts more than $\mathfrak{h} + \frac{(m-\mathfrak{h})\rho}{1-\rho}$ instances of $B$ to be negative. Then, since exactly $\mathfrak{h}$ instances of $B$ are negative in every distribution, the learner will predict more than $\frac{(m-\mathfrak{h})\rho}{1-\rho}$ positive instances of $B$ to be negative. Therefore, regardless of the distribution, its error will be more than $\rho$, which is greater than the error of always predicting positive. Thus, without loss of generality, we can assume that the learner always predicts fewer than $\mathfrak{h} + \frac{(m-\mathfrak{h})\rho}{1-\rho}$ negative labels. In the worst case, all of the negative predictions are over positive instances, in which case the error will be less than

$$\mathfrak{h} \cdot \frac{\rho}{\mathfrak{h}} + \left( \mathfrak{h} + \frac{(m-\mathfrak{h})\rho}{(1-\rho)} \right) \frac{1-\rho}{m-\mathfrak{h}} \leq \rho + 2\frac{(m-\mathfrak{h})\rho}{(1-\rho)} \cdot \frac{1-\rho}{m-\mathfrak{h}} = 3\rho$$

where the inequality is due to $m \geq \frac{\mathfrak{h}}{\rho} + \mathfrak{h}$. In conclusion, without loss of generality, we can assume that any learner always incurs an error less than $3\rho$.

Define $V = \{ O \subseteq B \mid |O| = \mathfrak{h} \}$. Note that $|V| = \binom{m}{\mathfrak{h}}$. Since maximum is no less than the average, we derive

$$\max_{O \in V} \mathbb{E}_{S^P \sim \mathcal{D}_{+,O}^b, S^U \sim \mathcal{D}_{\mathcal{X},O}^a} \left[ \text{err}_{\mathcal{D}_O} \left( \mathcal{A}(S^P, S^U) \right) \right]$$

$$\geq \frac{1}{\binom{m}{\mathfrak{h}}} \sum_{O \in V} \mathbb{E}_{S^P \sim \mathcal{D}_{+,O}^b, S^U \sim \mathcal{D}_{\mathcal{X},O}^a} \left[ \text{err}_{\mathcal{D}_O} \left( \mathcal{A}(S^P, S^U) \right) \right]$$

$$= \frac{1}{\binom{m}{\mathfrak{h}}(m-\mathfrak{h})^b} \sum_{O \in V} \sum_{S^P \in (B \setminus O)^b} \mathbb{E}_{S^U \sim \mathcal{D}_{\mathcal{X},O}^a} \left[ \text{err}_{\mathcal{D}_O} \left( \mathcal{A}(S^P, S^U) \right) \right] \quad (10)$$

$$> \frac{\binom{m-b}{\mathfrak{h}}}{\binom{m}{\mathfrak{h}}} \frac{1}{\binom{m-b}{\mathfrak{h}}m^b} \sum_{O \in V} \sum_{S^P \in (B \setminus O)^b} \mathbb{E}_{S^U \sim \mathcal{D}_{\mathcal{X},O}^a} \left[ \text{err}_{\mathcal{D}_O} \left( \mathcal{A}(S^P, S^U) \right) \right]$$

$$> \frac{1}{3\binom{m-b}{\mathfrak{h}}m^b} \sum_{O \in V} \sum_{S^P \in (B \setminus O)^b} \mathbb{E}_{S^U \sim \mathcal{D}_{\mathcal{X},O}^a} \left[ \text{err}_{\mathcal{D}_O} \left( \mathcal{A}(S^P, S^U) \right) \right]$$

Where the last line is due to the fact that since $m \geq \mathfrak{h}(b) + \mathfrak{h}$, we have

$$\frac{\binom{m-b}{\mathfrak{h}}}{\binom{m}{\mathfrak{h}}} > \left( \frac{m-b-\mathfrak{h}}{m-\mathfrak{h}} \right)^{\mathfrak{h}} = \left( \frac{\mathfrak{h}-1}{\mathfrak{h}} \right)^{\mathfrak{h}} \geq \frac{1}{e}$$

Next, for any $S^P \in B^b$ define $W(S^P) := \{O \subseteq B \setminus S^P \mid |O| = \mathfrak{h}\}$, and notice that since $|B \setminus S^P| \geq m - b$ we have $|W(S^P)| \geq \begin{pmatrix} m - b \\ \mathfrak{h} \end{pmatrix}$. Therefore, using the fact that the average is no less than the minimum, we derive

$$\frac{1}{3 \begin{pmatrix} m - b \\ \mathfrak{h} \end{pmatrix} m^b} \sum_{O \in V} \sum_{S^P \in (B \setminus O)^b} \mathbb{E}_{S^U \sim \mathcal{D}_{\mathcal{X},O}^a} \left[ \mathrm{err}_{\mathcal{D}_O} \left( \mathcal{A}(S^P, S^U) \right) \right]$$

$$= \frac{1}{3 \begin{pmatrix} m - b \\ \mathfrak{h} \end{pmatrix} m^b} \sum_{S^P \in B^b} \sum_{O \in W(S^P)} \mathbb{E}_{S^U \sim \mathcal{D}_{\mathcal{X},O}^a} \left[ \mathrm{err}_{\mathcal{D}_O} \left( \mathcal{A}(S^P, S^U) \right) \right] \quad (11)$$

$$\geq \frac{1}{3} \min_{S^P \in B^b} \mathbb{E}_{O \sim U_{W(S^P)}} \left[ \mathbb{E}_{S^U \sim \mathcal{D}_{\mathcal{X},O}^a} \left[ \mathrm{err}_{\mathcal{D}_O} \left( \mathcal{A}(S^P, S^U) \right) \right] \right]$$

We fix any $S^P \in O^b$. Furthermore, define event $E$ to be the event that $\left| S^U \mid O \right| \leq \frac{\mathfrak{h}}{2}$. The idea similar to Theorem 1 is that, since $E$ has a significant probability, we can lower bound $\mathbb{E}_{O \sim U_{W(S^P)}} \left[ \mathbb{E}_{S^U \sim \mathcal{D}_{\mathcal{X},O}^a} \left[ \mathrm{err}_{\mathcal{D}_O} \left( \mathcal{A}(S^P, S^U) \right) \right] \right]$ by conditioning it on event $E$ as

$$\mathbb{E}_{O \sim U_{W(S^P)}} \left[ \mathbb{E}_{S^U \sim \mathcal{D}_{\mathcal{X},O}^a} \left[ \mathrm{err}_{\mathcal{D}_O} \left( \mathcal{A}(S^P, S^U) \right) \right] \right]$$

$$\geq \mathbb{E}_{O \sim U_{W(S^P)}} \left[ \mathbb{E}_{S^U \sim \mathcal{D}_{\mathcal{X},O}^a} \left[ \mathrm{err}_{\mathcal{D}_O} \left( \mathcal{A}(S^P, S^U) \right) \mid E \right] \Pr_{S^U \sim \mathcal{D}_{\mathcal{X},O}^a} [E] \right] \quad (12)$$

Next, we try to lower bound $\mathbb{E}_{O \sim U_{W(S^P)}} \left[ \mathbb{E}_{S^U \sim \mathcal{D}_{\mathcal{X},O}^a} \left[ \mathrm{err}_{\mathcal{D}_O} \left( \mathcal{A}(S^P, S^U) \right) \mid E \right] \right]$. Fix any $O \in V$, notice that $\mathcal{D}_{\mathcal{X},O}(x) = \mathcal{D}_{\mathcal{X},O}(x')$ is the same for every $x, x' \in O$. Moreover, $\mathcal{D}_{\mathcal{X},O}(x) = \mathcal{D}_{\mathcal{X},O}(x')$ for every $x, x' \in B \setminus O$. Therefore, for every $S^U \in B^a$, and every $O, O' \in W(S^P)$ such that $S^U \mid O = S^U \mid O'$, we have

$$\Pr_{S \sim \mathcal{D}_{\mathcal{X},O'}^a} [S = S^U \mid E] = \Pr_{S \sim \mathcal{D}_{\mathcal{X},O}^a} [S = S^U \mid E].$$

Therefore, by fixing $S^U \in B^a$ and any $S'$ with $|S'| \leq \mathfrak{h}/2$ which represents $S^U \mid O$, we can define $P_{S^U, S'} := \Pr_{S \sim \mathcal{D}_{\mathcal{X},O}^a} [S = S^U \mid E]$.

This fact gives away the idea of grouping all the $O \in W(S^P)$ that have the same $S^U \mid O$ for a fix $S^U \in B^a$. Formally, we have

$$\mathbb{E}_{O \sim U_{W(S^P)}} \left[ \mathbb{E}_{S^U \sim \mathcal{D}_{\mathcal{X},O}^a} \left[ \mathrm{err}_{\mathcal{D}_O} \left( \mathcal{A}(S^P, S^U) \right) \mid E \right] \right]$$

$$= \frac{1}{|W(S^P)|} \sum_{O \in W(S^P)} \sum_{S^U \in B^a} \Pr_{S \sim \mathcal{D}_{\mathcal{X},O}^a} [S = S^U \mid E] \mathrm{err}_{\mathcal{D}_O} \left( \mathcal{A}(S^P, S^U) \right)$$

$$= \frac{1}{|W(S^P)|} \sum_{S^U \in B^a} \sum_{\substack{S' \in B^* \\ |S'| \leq \mathfrak{h}/2}} \sum_{\substack{O \in W(S^P) \\ S^U | O = S'}} \Pr_{S \sim \mathcal{D}_{\mathcal{X},O}^a} [S = S^U \mid E] \, \mathrm{err}_{\mathcal{D}_O} \left( \mathcal{A}(S^P, S^U) \right) \quad (13)$$

$$= \frac{1}{|W(S^P)|} \sum_{S^U \in B^a} \sum_{\substack{S' \in B^* \\ |S'| \leq \mathfrak{h}/2}} P_{S^U, S'} \sum_{\substack{O \in W(S^P) \\ S^U | O = S'}} \mathrm{err}_{\mathcal{D}_O} \left( \mathcal{A}(S^P, S^U) \right)$$

Next, fixing any $S^U \in B^U$ and any $S'$ with $|S'| \le \mathfrak{h}/2$. Note that since the error is always no less than the error restricted over $B \setminus (S^P \cup S^U)$, we derive

$$\sum_{\substack{O \in W(S^P) \\ S^U | O = S'}} \operatorname{err}_{\mathcal{D}_O}\left(\mathcal{A}(S^P, S^U)\right)$$

$$\ge \sum_{\substack{O \in W(S^P) \\ S^U | O = S'}} \sum_{x \in B \setminus (S^P \cup S^U)} \left(\frac{\rho}{\mathfrak{h}} \mathbb{1}\{\mathcal{A}(S^P, S^U)(x) = 1, x \in O\} + \frac{1 - \rho}{m - \mathfrak{h}} \mathbb{1}\{\mathcal{A}(S^P, S^U)(x) = 0, x \notin O\}\right)$$

$$= \sum_{x \in B \setminus (S^P \cup S^U)} \sum_{\substack{O \in W(S^P) \\ S^U | O = S'}} \left(\frac{\rho}{\mathfrak{h}} \mathbb{1}\{\mathcal{A}(S^P, S^U)(x) = 1, x \in O\} + \frac{1 - \rho}{m - \mathfrak{h}} \mathbb{1}\{\mathcal{A}(S^P, S^U)(x) = 0, x \notin O\}\right)$$

$$= \sum_{x \in B \setminus (S^P \cup S^U)} \sum_{\substack{O \in W(S^P) \\ S^U | O = S'}} \left(\frac{\rho}{\mathfrak{h}} \frac{|O \setminus S'|}{|B \setminus (S^P \cup S^U)|} \mathbb{1}\{\mathcal{A}(S^P, S^U)(x) = 1\}\right.$$

$$\left. + \frac{1 - \rho}{m - \mathfrak{h}} \frac{|B \setminus (S^P \cup S^U \cup O)|}{|B \setminus (S^P \cup S^U)|} \mathbb{1}\{\mathcal{A}(S^P, S^U)(x) = 0\}\right)$$

(14)

Where the last line is due to symmetry of $W(S^P)$. Next note that since $|S'| \le \frac{\mathfrak{h}}{2}$ we have $\frac{|O \setminus S'|}{\mathfrak{h}} \ge \frac{1}{2}$. Moreover, since $m \ge 2(b + a) + \mathfrak{h}$, we have

$$\frac{|B \setminus (S^P \cup S^U \cup O)|}{m - \mathfrak{h}} \ge \frac{m - b - a - \mathfrak{h}}{m - \mathfrak{h}} \ge \frac{1}{2}$$

Combining these facts with (14), we derive

$$= \sum_{x \in B \setminus (S^P \cup S^U)} \sum_{\substack{O \in W(S^P) \\ S^U | O = S'}} \left(\frac{\rho \mathbb{1}\{\mathcal{A}(S^P, S^U)(x) = 1\} + (1 - \rho)\mathbb{1}\{\mathcal{A}(S^P, S^U)(x) = 0\}}{2|B \setminus (S^P \cup S^U)|}\right)$$

$$\overset{(i)}{\ge} \sum_{x \in B \setminus (S^P \cup S^U)} \sum_{\substack{O \in W(S^P) \\ S^U | O = S'}} \frac{\rho}{2}$$

(15)

$$= \sum_{\substack{O \in W(S^P) \\ S^U | O = S'}} \frac{\rho}{2}$$

Where (i) is due to the fact that $\rho \ge \frac{1}{2}$. Combining (13) and (15) we derive

$$\mathbb{E}_{O \sim U_{W(S^P)}}\left[\mathbb{E}_{S^U \sim \mathcal{D}_{\mathcal{X},O}^a}\left[\operatorname{err}_{\mathcal{D}_O}\left(\mathcal{A}(S^P, S^U)\right) \mid E\right]\right]$$

$$\ge \frac{1}{|W(S^P)|} \sum_{S^U \in B^a} \sum_{\substack{S' \in B^* \\ |S'| \le \mathfrak{h}/2}} P_{S^U, S'} \sum_{\substack{O \in W(S^P) \\ S^U | O = S'}} \frac{\rho}{2}$$

(16)

$$= \frac{1}{|W(S^P)|} \sum_{O \in W(S^P)} \sum_{S^U \in B^a} \Pr_{S \sim \mathcal{D}_{\mathcal{X},O}^a}[S = S^U \mid E]\frac{\rho}{2}$$

$$= \mathbb{E}_{O \sim U_{W(S^P)}}\left[\mathbb{E}_{S^U \sim \mathcal{D}_{\mathcal{X},O}^a}\left[\frac{\rho}{2} \mid E\right]\right] = \frac{\rho}{2}$$

Next, we attempt to lower bound $\Pr_{S^U \sim \mathcal{D}_{\mathcal{X},O}^a}[E]$. Similar to the proof of Theorem 1, using Multiplicative Chernoff bound (Lemma 21) for any $\gamma \in \left[0, \frac{\mathfrak{h}}{\rho} - 1\right]$ we derive

$$\Pr_{S^U \sim \mathcal{D}_{\mathcal{X},O}^a}\left[|S^U \mid O| \ge (1 + \gamma)a\rho\right] \le e^{-\frac{a\rho\gamma^2}{3}}$$

We set $\gamma = 1$. Note that since $\varepsilon = \min\left[\frac{7}{2400}, \frac{7\hbar}{4800a}\right]$, we get $\rho = \min\left[\frac{1}{2}, \frac{\hbar}{4a}\right]$. Thus, we get

$$\Pr_{S^U \sim \mathcal{D}^a_{\mathcal{X},O}}\left[|S^U \mid O| \geq \frac{\hbar}{2}\right] \leq e^{-\frac{\hbar}{12}} < 0.93.$$

Therefore, $\Pr_{S^U \sim \mathcal{D}^a_{\mathcal{X},O}}[E] > 0.07$. Combing this fact with (10), (11), (12) and (16) we derive

$$\max_{O \in V} \mathbb{E}_{S^P \sim \mathcal{D}^b_{+,O}, S^U \sim \mathcal{D}^a_{\mathcal{X},O}}\left[\mathrm{err}_{\mathcal{D}_O}\left(\mathcal{A}(S^P, S^U)\right)\right] > \frac{7}{600}\rho$$

Thus, there exists a $O^\star \in V$ such that $\mathbb{E}_{S^P \sim \mathcal{D}^b_{+,O^\star}, S^U \sim \mathcal{D}^a_{\mathcal{X},O^\star}}\left[\mathrm{err}_{\mathcal{D}_{O^\star}}\left(\mathcal{A}(S)\right)\right] > \frac{7}{600}\rho$. Since the error is always less than $3\rho$, and by using Lemma 20 we derive

$$\Pr_{S^P \sim \mathcal{D}^b_{+,O^\star}, S^U \sim \mathcal{D}^a_{\mathcal{X},O^\star}}\left[\mathrm{err}_{\mathcal{D}_{O^\star}}\left(\mathcal{A}(S)\right) \geq \varepsilon\right] \geq \frac{\frac{7}{600}\rho - \varepsilon}{(3\rho - \varepsilon)}$$

$$= \frac{\varepsilon}{\frac{3600}{7}\varepsilon - \varepsilon}$$

$$= \frac{7}{3599} > \frac{1}{1000}.$$

and this completes the proof.

*Proof of $m^{unlabel}_{\mathcal{C}}(\varepsilon, \delta) = \Omega\left(\frac{\ln(1/\delta)}{\varepsilon}\right)$.* Next we prove that for every $b \in \mathbb{N}$, $\varepsilon < \frac{1}{8}$, $\delta < \frac{1}{4}$, $a = \frac{\ln\left(\frac{1}{4\delta}\right)}{2\varepsilon}$ and PU learner $\mathcal{A}$ there is a distribution $\mathcal{D}$ realized by $\mathcal{C}$ over $\mathcal{X} \times \{0, 1\}$ such that

$$\Pr_{S^P \sim \mathcal{D}^b_+, S^U \sim \mathcal{D}^a_{\mathcal{X}}}\left[\mathrm{err}_{\mathcal{D}}\left(\mathcal{A}(S^P, S^U)\right) \geq \varepsilon\right] > \delta.$$

Let $m := 2(b + a + 1)$. Note that since claw number is more than 1, there exists a $B \subseteq \mathcal{X}$ with size $m$ such that $B \setminus \{x\} \subseteq \mathcal{C}$ for all $x \in B$. For any $x \subseteq B$ and any $(x', y') \in \mathcal{X} \times \{0, 1\}$ we define the distribution $\mathcal{D}_x$ over $\mathcal{X} \times \{0, 1\}$ as

$$\mathcal{D}_x(\{(x', y')\}) := \begin{cases} \varepsilon & \text{if } x' = x, \text{ and } y = 0 \\ \frac{(1-\varepsilon)}{(m-1)} & x' \in B \setminus \{x\}, \text{ and } y = 1 \\ 0 & \text{o.w.} \end{cases} \tag{17}$$

Note that all such $\mathcal{D}_x$ are realized by $\mathcal{C}$. It is enough to show that

$$\Pr_{x^\star \sim U_B, S^P \sim \mathcal{D}^b_{+,x^\star}, S^U \sim \mathcal{D}^a_{\mathcal{X},x^\star}}\left[\mathrm{err}_{\mathcal{D}}\left(\mathcal{A}(S^P, S^U)\right) \geq \varepsilon\right] > \delta.$$

Fix any $x^\star \in B$. Note that similar to the manner (7) in the proof of Theorem 9 was obtained, for all $\varepsilon < \frac{1}{2}$ we derive

$$\Pr_{S^U \sim \mathcal{D}^a_{\mathcal{X},x^\star}}[x^\star \notin S^U] = (1 - \varepsilon)^a > e^{-2\varepsilon a} = 4\delta. \tag{18}$$

Next, consider any positive sample $S^P$ and unlabeled sample $S^U$ such that $S^U$ doesn't contain $x^\star$. Note that the probability of $S^P$ and $S^U$ is the same w.r.t. every $\mathcal{D}_x$ such that $x \in B \setminus (S^U \cup S^P)$. We consider two cases based on the number of negative labels $\mathcal{A}(S^P, S^U)$ predicts. We prove that for both cases the error of $\mathcal{A}(S^P, S^U)$ is more than $\varepsilon$ with probability at least $\frac{1}{4}$. Therefore, the probability that error of $\mathcal{A}(S^P, S^U)$ is at least $\epsilon$ would be more than

$$\frac{\Pr_{S^U \sim \mathcal{D}^a_{\mathcal{X},x^\star}}[x^\star \notin S^U]}{4} = \delta,$$

which completes the proof.

Case 1: $\mathcal{A}(S^P, S^U)$ has more than $2m\varepsilon + 1$ negative labels. Then, at least $2m\varepsilon$ of them are not $x^\star$, and subsequently, since $\varepsilon < \frac{1}{2}$ the error is guaranteed to be at least

$$\frac{2m\varepsilon(1 - \varepsilon)}{m - 1} > \varepsilon.$$

Case 2: $\mathcal{A}(S^P, S^U)$ at most $2m\varepsilon + 1$ negative labels. Note that, since $m = 2(b + a + 1)$, we have $|B \setminus (S^U \cup S^P)| \geq \frac{m}{2} + 1$. Therefore, since $\varepsilon < \frac{1}{8}$, $\mathcal{A}(S^P, S^U)$ predicts at least $m\left(\frac{1}{2} - 2\varepsilon\right) \geq \frac{m}{4}$ positive labels over $B \setminus (S^U \cup S^P)$. Therefore, since $x^\star$ is drawn uniformly, with more than $\frac{1}{4}$ chance the label of $x^\star$ would be predicted positive, which indicates that $\mathcal{A}(S^P, S^U)$ has more than $\varepsilon$ error.

$\square$

**Proposition 26.** *For a concept class $\mathcal{C}$, let $\mathcal{C}_\cap := \left\{\bigcap_{c \in A} c \mid \text{finite } A \subseteq \mathcal{C}\right\}$. Then $\mathrm{VCD}(\mathcal{C}_\cap) = \infty$ if and only if the claw number of $\mathcal{C}$ is at least 1.*

*Proof. Only if side.* Since $\mathrm{VCD}(\mathcal{C}_\cap) = \infty$, for every $m \in \mathbb{N}$, there exists a subset $B = \{x_1, ..., x_m\}$ that is shattered by $\mathcal{C}_\cap$. Thus, for all $i \in [m]$ there should be a $c_\cap \in \mathcal{C}_\cap$ such that $c_\cap \cap B = B \setminus \{x_i\}$. This indicates that $B \setminus \{x_i\} \subseteq c_\cap$. Consequently, there should be a $c \in \mathcal{C}$ such that $c_\cap \subseteq c$, and $x_i \notin c$. Since $B \setminus \{x_i\} \subseteq c$, in turn, implies that $c \cap B = B \setminus \{x_i\}$. Thus, $\{O \subseteq B \mid |O| = m - 1\} \subseteq \mathcal{C} \mid B$, and therefore the claw number of $\mathcal{C}$ is at least 1.

*If side:* Since claw number is at least 1, for every $m \in \mathbb{N}$ there exists a subset $B = \{x_1, ..., x_{m+1}\}$ such that for all $i \in [m + 1]$ we have $B \setminus \{x_i\} \in \mathcal{C} \mid B$. For every $i \in [m + 1]$ define $c_i$ to be the concept such that $c_i \cap B = B \setminus \{x_i\}$.

Denote $\bar{B} := B \setminus \{x_{m+1}\}$. Next, consider any $A \subseteq \bar{B}$ we prove that there exists a $c_\cap \in \mathcal{C}_\cap$ that satisfies $c_\cap \cap \bar{B} = A$, and this shows that $\bar{B}$ is shattered by $\mathcal{C}_\cap$. This implies that for every $m \in \mathbb{N}$ we have $\mathrm{VCD}(\mathcal{C}_\cap) \geq m$, which completes the proof. When $A = \bar{B}$ define $c_\cap := c_{m+1}$, and we derive $c_\cap \cap \bar{B} = \bar{B}$. Otherwise, define $c_\cap := \bigcap_{i : x_i \in \bar{B} \setminus A} c_i$. It is easy to see $c_\cap$ satisfies the desired property. $\square$

**Lemma 27.** *[Corollary 5 of Liu et al. (2002)] Let $\mathcal{C}$ be a concept class with VC-dimension $d$ over $\mathcal{X}$. Let $S$ be an i.i.d. sample of size $n$ from a distribution $\mathcal{D}_\mathcal{X}$ over $\mathcal{X}$. There exists a constant $M > 1$, such that if $n > M\left(\frac{d \ln(1/\varepsilon) + \ln(1/\delta)}{\varepsilon}\right)$, then for every $c, c^\star \in \mathcal{C}$ with probability $1 - \delta$ we have*

$$Pr\left(c(x) \neq c^\star(x)\right) > \frac{3 \sum_{x \in S} \mathbb{1}\{c(x) \neq c^\star(x)\}}{n} + \epsilon$$

*and*

$$\frac{\sum_{x \in S} \mathbb{1}\{c(x) \neq c^\star(x)\}}{n} > 3 Pr\left(c(x) \neq c^\star(x)\right) + \epsilon$$

**Lemma 28.** *Let $\mathcal{C}$ be a realizable concept class with VC dimension $d$ over domain $\mathcal{X}$. Let $S$ be a sample i.i.d. drawn from $\mathcal{D}_\mathcal{X}$ and $T \in \mathcal{X}^*$ be an $\varepsilon-$net for $\mathcal{C} \triangle \mathcal{C}$ on $\mathcal{D}_+$ such that $Domain(T) \subseteq c^\star$. Denote $c^{PU} := \mathrm{argmin}_{c \in \mathcal{C}, Domain(T) \subseteq c} |c \mid S|$. Then there exists a $M > 1$ such that if $|S| > M\left(\frac{d \ln(1/\varepsilon) + \ln(1/\delta)}{\varepsilon}\right)$, then with probability $1 - 2\delta$ we have $\mathrm{err}_\mathcal{D}(c^{PU}) \leq 14\varepsilon$.*

*Proof.* Proof of this theorem was extracted from Theorem 1 of Liu et al. (2002). First, note that according to the definition of $\varepsilon-$net, for any $c$ such that $T \subseteq c$ we have that

$$\Pr(c(x) = 0, y = 1) \leq \mathrm{err}_\mathcal{D}^P(c) \leq \varepsilon. \tag{19}$$

Note that since $T \subseteq c^\star$, we have $|c^{PU} \mid S| \leq |c^\star \mid S|$. Thus,

$$|c^{PU} \cap c^\star \mid S| + |c^{PU} \cap \overline{c^\star} \mid S| \leq |c^\star \mid S|$$

Therefore,

$$|c^{PU} \cap \overline{c^\star} \mid S| \leq |c^\star \mid S| - |c^\star \cap c^{PU} \mid S| \tag{20}$$
$$= |\overline{c^{PU}} \cap c^\star \mid S|$$

From Lemma 27 it can be deduced that there exists a $M > 1$ such that as long as $|S| \geq M\left(\frac{d \ln(1/\varepsilon) + \ln(1/\delta)}{\varepsilon}\right)$ with probability $1 - \delta$ we have

$$\frac{|\overline{c^{PU}} \cap c^\star \mid S|}{|S|} \leq 3 Pr(c^\star(x) = 1, c^{PU}(x) = 0) + \varepsilon \tag{21}$$

Similarly as long as $|S| \geq M\left(\frac{d \ln(1/\varepsilon)+\ln(1/\delta)}{\varepsilon}\right)$ with probability $1-\delta$ we have

$$\Pr(c^\star(x)=0, c^{PU}(x)=1)) \leq 3\frac{|c^{PU} \cap \overline{c^\star} \mid S|}{|S|} + \varepsilon \tag{22}$$

Combining (20), (21) and (22) with probability $1-2\delta$ we have

$$\Pr(c^\star(x)=0, c^{PU}(x)=1) \leq 9\Pr(c^\star(x)=0, c^{PU}(x)=1) + 4\varepsilon$$
$$\overset{(i)}{\leq} 13\varepsilon \tag{23}$$

Where (i) is due to (19) and $\Pr(c^\star(x) \neq y) = 0$. Thus combining (19) and (23) and the fact that $\Pr(c^\star(x) \neq y) = 0$ with probability $1-2\delta$ we derive

$$\text{err}_{\mathcal{D}}(c^{PU}) = \Pr(c^{PU}(x) \neq c^\star(x))$$
$$= \Pr(c^\star(x)=0, c^{PU}(x)=1) + \Pr(c^\star(x)=0, c^{PU}(x)=1)$$
$$\leq 14\varepsilon.$$

Which completes the proof. $\qquad\square$

## C   Missing Proofs from Section 4

**Theorem 8.** *Let $\mathcal{C}$ be a concept class over domain $\mathcal{X}$ with VC dimension $d$ and $r \in (0,1)$. Let $\mathcal{W}$ be a set of duos $(\mathcal{P}, \mathcal{D})$ such that $\mathcal{D}$ is realized by $\mathcal{C}$, $\mathcal{P} \in \mathcal{K}_{\mathcal{D}}^{sar}$, and $R_{\mathcal{C}\Delta\mathcal{C}}(\mathcal{P}, \mathcal{D}_+) \geq r$. Then PERM algorithm (1) PU learns $\mathcal{C}$ over $\mathcal{W}$ with sample complexity $m_{\mathcal{C}}^{unlabel}(\varepsilon, \delta) = O\left(\frac{d\ln(1/\varepsilon)+\ln(1/\delta)}{\varepsilon}\right)$ and $m_{\mathcal{C}}^{pos}(\varepsilon, \delta) = O\left(\frac{d\ln(1/r\varepsilon)+\ln(1/\delta)}{r\varepsilon}\right)$.*

*Proof.* As mentioned in Corollary 6, it is well-known that as long as $b > M\left(\frac{d\ln(1/r\varepsilon)+\ln(1/\delta)}{r\varepsilon}\right)$ for a constant $M$, $S^P$ is a $r\varepsilon-$net for $\mathcal{C}\Delta\mathcal{C}$ with respect to $\mathcal{P}$ with probability $1-\delta$. Using lemma 7 we derive that $S^P$ is a $\varepsilon-$net for $\mathcal{C}\Delta\mathcal{C}$ with respect to $\mathcal{D}_+$ with probability $1-\delta$. Moreover, we know $\mathcal{P}(A)=0$ for every measurable $A$ that has $\mathcal{D}_+(A)=0$. This indicates that given any $c^\star$ with $\text{err}_{\mathcal{D}}(c^\star)=0$, almost surely we have $S^P \subseteq c^\star$. Plugging these into lemma 5 completes the proof.

$\qquad\square$

**Theorem 9.** *Let $\mathcal{C}$ be a concept class over domain $\mathcal{X}$ with VC dimension $d \geq 2$ and $r \in (0,1)$. There exists a $M > 1$ such that for any number of positive samples upper bounded by $b \leq M\left(\frac{d+\ln(1/\delta)}{r\varepsilon}\right)$ and any number of unlabeled samples $a \in \mathbb{N}$, $\varepsilon, \delta \in (0,1) \times (0,1)$, and PU learner $\mathcal{A}$, there is a distribution $\mathcal{D}$ realized by $\mathcal{C}$ over $\mathcal{X} \times \{0,1\}$ and a distribution $\mathcal{P} \in \mathcal{K}_{\mathcal{D}}^{sar}$ such that $R(\mathcal{P}, \mathcal{D}_+) \geq r$ and $\Pr_{S^P \sim \mathcal{P}^b, S^U \sim \mathcal{D}_{\mathcal{X}}^a}\left[\text{err}_{\mathcal{D}}(\mathcal{A}(S^P, S^U)) \geq \varepsilon\right] > \delta$.*

*Proof.* Let $\rho = 320\varepsilon$, $B = \{x_1, ..., x_d\}$ be any set of size $d$ shattered by $\mathcal{C}$, and denote $\bar{B} := B \setminus \{x_d\}$. Consider the set of distributions $\mathcal{W}_{\rho,d}^{scar-pos} = \{(\mathcal{D}_O, \mathcal{D}_{+,O}) : O \subseteq \bar{B}\}$, as defined in the proof of Theorem 1. For each $O \subseteq \bar{B}$, define a distribution $\mathcal{P}_O$ such that $\mathcal{P}_O(\{(x,y)\}) := r \cdot \mathcal{D}_{+,O}(\{(x,y)\})$ for every $x \in \bar{B}$ and $y \in \{0,1\}$, and assign the remaining probability mass of $\mathcal{P}_O$ to the point $(x_d, 1)$. By construction, we have $R(\mathcal{P}_O, \mathcal{D}_{+,O}) \geq r$.

Now define the set of duos $\mathcal{W}_{\rho,d}^{sar} := \{(\mathcal{P}_O, \mathcal{D}_O) \mid O \subseteq \bar{B}\}$. By denoting $\rho' := 320\varepsilon r$, it is easy to see that PU learnability over $\mathcal{W}_{\rho,d}^{sar}$ is equivalent with PU learnability over $\mathcal{W}_{\rho',d}^{scar-pos}$. Consequently, from the argument in the proof of Theorem 1, for $d \geq 9$ the number of positive examples required by any algorithm that PU learns $\mathcal{C}$ over $\mathcal{W}_{\rho,d}^{sar}$ must satisfy

$$m_{\mathcal{C}}^{pos}(\varepsilon, \tfrac{1}{319}) \geq \frac{d-1}{512\varepsilon r}.$$

For $d \geq 2$ an identical argument also gives

$$m_{\mathcal{C}}^{pos}(\varepsilon, \delta) \geq \frac{\ln(1/2\delta)}{2r\varepsilon}.$$

This completes the proof. $\qquad\square$

**Theorem 10.** *Consider any finite domain $\mathcal{X}$. There exists a concept class $\mathcal{C}_{0,1}$ with $\mathrm{VCD}(\mathcal{C}_{0,1}) = 1$, such that for every PU learner $\mathcal{A}$, and $\varepsilon$ and $\delta$ with $2\varepsilon + \delta < 1/2$, $b, a \in \mathbb{N}$ such that the total number of positive and unlabeled data is upper bounded by $b + a < \sqrt{\frac{2(1-2(2\varepsilon+\delta))|\mathcal{X}|}{3}} - 2$, there exists a distribution $\mathcal{D}$ over $\mathcal{X} \times \{0, 1\}$ with deterministic labels which is realized by $\mathcal{C}_{0,1}$ and $\mathcal{P} \in \mathcal{K}_{\mathcal{D}}^{cov}$ where $R(\mathcal{P}, \mathcal{D}_+) = 1/2$, $\mathcal{D}[\{(x, 1) : x \in \mathcal{X}\}] \geq 1/2$ and $\Pr_{S^P \sim \mathcal{P}^b, S^U \sim \mathcal{D}_{\mathcal{X}}^a}\left[\mathrm{err}_{\mathcal{D}}(\mathcal{A}(S^P, S^U) \geq \varepsilon\right] > \delta$.*

We obtain our lower bound by reducing the following problem to the PU learning problem:

**The Left/Right Problem.** We consider the problem of distinguishing two distributions from finite samples. The Left/Right Problem was introduced in Kelly et al. (2010):

**Input:** Three finite samples, $L$, $R$ and $M$ of points from some domain set $\mathcal{X}$.

**Output:** Assuming that $L$ is an i.i.d. sample from some distribution $P$ over $\mathcal{X}$, that $R$ is an i.i.d. sample from some distribution $Q$ over $\mathcal{X}$, and that $M$ is an i.i.d. sample generated by one of these two probability distributions, was $M$ generated by $P$ or by $Q$ ?

*Lower bound for the Left/Right problem*: We say that a (randomized) algorithm $(\delta, l, r, m)$-solves the Left/Right problem if, given samples $L$, $R$ and $M$ of sizes $l$, $r$ and $m$ respectively, it gives the correct answer with probability at least $1 - \delta$. Lemma 1 of Ben-David and Urner (2012) shows that for any sample sizes $l$, $r$ and $m$ and for any $\gamma < 1/2$, there exists a finite domain $\mathcal{X} = \{1, 2, \ldots, n\}$ and a finite class $\mathcal{W}_n^{uni}$ of triples of distributions over $\mathcal{X}$ such that no algorithm can $(\gamma, l, r, m)$-solve the Left/Right problem for this class. In this class, both the distribution generating $L$ and the distribution generating $R$ are uniform over half of the points in $\mathcal{X}$, but their supports are disjoint. More formally,

$$\mathcal{W}_n^{uni} = \{(U_A, U_B, U_C) : A \cup B = \{1, \ldots, n\}, A \cap B = \emptyset, |A| = |B|, \text{ and } C = A \text{ or } C = B\},$$

where, for a finite set $Y$, $U_Y$ denotes the uniform distribution over $Y$.

**Lemma 29** (Lemma 1 of Ben-David and Urner (2012))**.** *For any given sample sizes $l$ for $L$, $r$ for $R$ and $m$ for $M$ and any $0 < \gamma < 1/2$, if $k = \max\{l, r\} + m$, then for $n > (k + 1)^2/(1 - 2\gamma)$ no algorithm has a probability of success greater than $1 - \gamma$ over the class $\mathcal{W}_n^{uni}$*

*Reducing the Left/Right problem to PU learning:* In order to reduce the Left/Right problem to PU learning, we define a class of PU learning problems that corresponds to the class of duos $\mathcal{W}_n^{uni}$, for which we have proven a lower bound on the sample sizes needed for solving the Left/Right problem. Let $\mathcal{X}$ be some domain of size $n$. Let $Y$ be the first $n/3$ elements of $\mathcal{X}$ and $Z$ to be the next $2n/3$ elements of it. Let $\mathcal{C}_{0,1} = \{c_0, c_1\}$ where $c_1 = \mathcal{X}$ and $c_0 = Z$. Clearly $\mathrm{VCD}(\mathcal{C}_{0,1}) = 1$. We define $\mathcal{W}_n^{cov}$ to be the class of duos $(\mathcal{P}, \mathcal{D})$, where $\mathcal{D}$ is a distribution with a deterministic label such that $\mathcal{D}_{\mathcal{X}}$ is uniform either over $Y \cup J$ or $Y \cup (Z \backslash J)$ for some uniform subset $J$ of $Z$ of size $n/3$, and $l$ assigns points in $Y \cup J$ to 1 and points in $Z \backslash J$ to 0, and $\mathcal{P}$ is uniform over $Y \cup J$. Notice that $\mathcal{P} \in \mathcal{K}_{\mathcal{D}}^{cov}$. Note that we always either have $R(\mathcal{P}, \mathcal{D}_+) = 1/2$ or $R(\mathcal{P}, \mathcal{D}_+) = 1$. Moreover, since the label of $Y$ is always 1, $\mathcal{D}[\{(x, 1) : x \in \mathcal{X}\}] \geq 1/2$. Furthermore, for all $(\mathcal{P}, \mathcal{D})$ in $\mathcal{W}_n^{cov}$

$$\min_{c \in \mathcal{C}_{0,1}} \mathrm{err}_{\mathcal{D}}(c) = 0$$

for all elements of $\mathcal{W}_n^{cov}$.

**Lemma 30.** *Consider any $s, t \in \mathbb{N}$. The Left/Right problem reduces to Domain Adaptation. More precisely, given a number $n$, suppose there exists a PU learner $\mathcal{A}$ such that for all $(\mathcal{P}, \mathcal{D}) \in \mathcal{W}_{3n/2}^{cov}$, $b \geq s$ and $a \geq t$ satisfies*

$$\mathbb{E}_{S^P \sim \mathcal{P}^b, S^U \sim \mathcal{D}_{\mathcal{X}}^a}\left[\mathrm{err}_{\mathcal{D}}(\mathcal{A}(S^P, S^U) \geq \varepsilon\right] \leq \delta.$$

*Then, we can construct an algorithm that $(2\varepsilon + \delta, s, s, t + 1)$-solves the Left/Right problem on $\mathcal{W}_n^{uni}$.*

*Proof.* Assume we are given samples $L = \{l_1, l_2, \ldots, l_s\}$ and $R = \{r_1, r_2, \ldots, r_s\}$ of size $s$ and a sample $M$ of size $t + 1$ for the Left/Right problem coming from a triple $(U_A, U_B, U_C)$ of distributions in $\mathcal{W}_n^{uni}$. Then consider any set $Y$ of size $n/2$ distinct from $A$ and $B$. We create the set $I^U$ in the following manner. Initiate $I^U = \emptyset$. Keep flipping an unbiased coin until $t$ of them are head, and each time the outcome of the coin was tail sample a $z \sim U_Y$ and add it to $I^U$. Similarly, initiate $I^P = \emptyset$, and keep flipping an unbiased coin until $s$ of them are head, and each time the outcome of the coin was tail sample a $z \sim U_Y$ and add it to $I^P$.

We construct an input to PU learning problem by setting the unlabeled sample $S^U = M\backslash\{p\} \cup I^U$, where $p$ is a point from $M$ chosen uniformly at random, and setting the positive sample $S^P = R \cup I^P$. Observe that $|S^P| \geq s$ and $|S^U| \geq t$. These sets can now be considered as an input to the PU learning problem generated from a sampling positive distribution $\mathcal{P} = U_{Y \cup B}$, and distribution $\mathcal{D}$ such that $\mathcal{D}_{\mathcal{X}}$ equal to $U_{Y \cup A}$ or to $U_{Y \cup B}$ (depending on whether $M$ was a sample from $U_A$ or from $U_B$) with labeling function being $l(x) = 0$ if $x \in A$ and $l(x) = 1$ if $x \in Y \cup B$. Observe that we have $R(\mathcal{P}, \mathcal{D}_+) = 1/2$ or $R(\mathcal{P}, \mathcal{D}_+) = 1$, and $\min_{c \in \mathcal{C}_{0,1}} \text{err}_{\mathcal{D}}(c) = 0$, and $(\mathcal{P}, \mathcal{D}) \in \mathcal{W}_{3n/2}^{cov}$. Denote $c = \mathcal{A}(S^P, S^U)$. The algorithm for the Left/Right problem then outputs $U_A$ if $c(p) = 0$ and $U_B$ if $c(p) = 1$. Note that $\text{err}_{\mathcal{D}}(c) \leq \varepsilon$ with confidence $1 - \delta$. Thus, $\text{err}_{U_M} \leq 2\varepsilon$ with confidence $1 - \delta$. Therefore, the algorithm is correct with probability at least $2\varepsilon + \delta$.

$\square$

*Proof of Theorem 10.* Combining Lemma 30 and Lemma 29 we conclude that there exists no $\mathcal{A}$ such that for all $(\mathcal{P}, \mathcal{D}) \in \mathcal{W}_{3n/2}^{cov}$ satisfies

$$\mathbb{E}_{S^P \sim \mathcal{P}^b, S^U \sim \mathcal{D}_{\mathcal{X}}^a} \left[ \text{err}_{\mathcal{D}}(\mathcal{A}(S^P, S^U)) \geq \varepsilon \right] \leq \delta,$$

unless $b + a \geq \sqrt{(1 - 2(2\varepsilon + \delta))n} - 2$, which completes the proof.

**Theorem 11.** *Let $\mathcal{X} = [0,1]^k$. There exists a concept class $\mathcal{C}_{0,1}$ with $\text{VCD}(\mathcal{C}_{0,1}) = 1$, such that for every PU learner $\mathcal{A}$, and $\varepsilon$ and $\delta$ with $2\varepsilon + \delta < 1/2$, $b, a \in \mathbb{N}$ such that the total number of positive and unlabeled data is upper bounded by $b + a < \sqrt{\frac{2(1+\lambda)^k(1 - 2(2\varepsilon + \delta))}{3}} - 2$, there exists a distribution $\mathcal{D}$ over $\mathcal{X} \times \{0,1\}$ with deterministic labels which is realized by $\mathcal{C}_{0,1}$ and $\mathcal{P} \in \mathcal{K}_{\mathcal{D}}^{cov}$ where $C(\mathcal{P}, \mathcal{D}_+) = 1/2$, $\mathcal{D}[\{(x, 1) : x \in \mathcal{X}\}] \geq 1/2$ and $l$ is a $\lambda$-Lipschitz labeling function and $\Pr_{S^P \sim \mathcal{P}^b, S^U \sim \mathcal{D}_{\mathcal{X}}^a} \left[ \text{err}_{\mathcal{D}}(\mathcal{A}(S^P, S^U)) \geq \varepsilon \right] > \delta$.*

*Proof.* Let $\mathcal{J} \subseteq \mathcal{X}$ be the points of a grid in $[0,1]^k$ with distance $1/\lambda$. Then we have $|\mathcal{J}| = (\lambda + 1)^k$. Then the class $\mathcal{W}_\lambda$ contains all duos $(\mathcal{P}, \mathcal{D})$, where the support of $\mathcal{P}$ and $\mathcal{D}$ is $\mathcal{J}$, $\mathcal{D}$ has deterministic labels and is realized by $\mathcal{C}_{0,1}$ and $\mathcal{P} \in \mathcal{K}_{\mathcal{D}}^{cov}$, $C(\mathcal{P}, \mathcal{D}_+) = 1/2$, and $\mathcal{D}[\{(x, 1) : x \in \mathcal{X}\}] \geq 1/2$, with any arbitrary labeling functions $l : \mathcal{J} \to \{0,1\}$, as every such function is $\lambda$-Lipschitz. As $\mathcal{J}$ is finite, the bound follows from Theorem 10. $\square$

**Theorem 12.** *Let $\mathcal{X} = [0,1]^k, \gamma > 0$ a margin parameter, $\pi, r > 0$ and $\mathcal{C}$ be a realizable concept class with VC dimension $d < \infty$. Let $\mathcal{W}$ to be the set of duos $(\mathcal{P}, \mathcal{D})$ such that:*

- *$\mathcal{P} \in \mathcal{K}_{\mathcal{D}}^{cov}$, $\mathcal{D}$ is realizable by $\mathcal{C}$ with margin $\gamma$ and has deterministic labels, and $\mathcal{D}(y = 1) \geq \pi$.*

- *The labeling function $l$ is a $\gamma$-margin classifier with respect to $\mathcal{D}_{\mathcal{X}}$.*

- *$R_{\mathcal{I}}(\mathcal{P}, \mathcal{D}_+) \geq r$ for the class $\mathcal{I} = (\mathcal{C}\Delta\mathcal{C}) \sqcap \mathcal{B}$, where $\mathcal{B}$ is a partition of $[0,1]^k$ into boxes of sidelength $\gamma/\sqrt{k}$.*

*Then Algorithm 1 PU learns $\mathcal{C}$ over $\mathcal{W}$ with sample complexity*

$$m_{\mathcal{C}}^{pos}(\varepsilon, \delta) = O\left( \frac{d \ln(1/(r(1-\varepsilon)\varepsilon)) + \ln(1/\delta)}{r^2(1-\varepsilon)^2\varepsilon} \right),$$

$$m_{\mathcal{C}}^{unlabel}(\varepsilon, \delta) = O\left( \frac{(\sqrt{k}/\gamma)^k \ln\left((\sqrt{k}/\gamma)^k/\delta\right)}{\pi\varepsilon} + \frac{d\ln(1/\varepsilon) + \ln(1/\delta)}{\varepsilon} \right).$$

*Proof.* Let $\varepsilon > 0$ and $\delta > 0$ be given. We set $\varepsilon' = \varepsilon/28$ and $\delta' = \delta/6$ and divide the space $\mathcal{X}$ up into heavy and light boxes from $\mathcal{B}$, by defining a box $A \in \mathcal{B}$ to be light if $\mathcal{D}_+(A) \leq \varepsilon'/|\mathcal{B}| = \varepsilon'/(\sqrt{k}/\gamma)^k$ and heavy otherwise. We let $\mathcal{X}^l$ denote the union of the light boxes and $\mathcal{X}^h$ the union of the heavy boxes. Further, we let $\mathcal{P}_h$ and $\mathcal{D}_{+,h}$ denote the restrictions of the $\mathcal{P}$ and $\mathcal{D}_+$ to $\mathcal{X}^h$, i.e. we have $\mathcal{P}_h(A) = \mathcal{P}(A)/\mathcal{P}(\mathcal{X}^h)$ and $\mathcal{D}_{+,h}(A) = \mathcal{D}_+(A)/\mathcal{D}_+(\mathcal{X}^h)$ for all $A \subseteq \mathcal{X}^h$ and $\mathcal{P}_h(A) = \mathcal{D}_{+,h}(A) = 0$ for all $A \not\subseteq \mathcal{X}^h$. As $|\mathcal{B}| = (\sqrt{k}/\gamma)^k$, we have

$\mathcal{D}_+\left(\mathcal{X}^h\right) \geq 1 - \varepsilon'$ and thus, $\mathcal{P}\left(\mathcal{X}^h\right) \geq r\left(1 - \varepsilon'\right)$. We will show that

**Claim 1.** With probability $1 - \delta'$ we have $S^U$ hits every heavy box (Similar to claim 1 of Theorem 3 of Ben-David and Urner (2012)).

**Claim 2.** With probability at least $1 - 2\delta'$ the intersection of $S^P$ and $\mathcal{X}^h$ is an $\varepsilon'$-net for $\mathcal{C}\Delta\mathcal{C}$ with respect to $\mathcal{D}_{+,h}$ (claim 2 of Theorem 3 of Ben-David and Urner (2012)).

To see that these imply the claim of the theorem, let $S^h = S^P \mid \mathcal{X}^h$ denote the intersection of the source sample and the union of heavy boxes. By Claim 1, $S^U$ hits every heavy box with high probability, thus $S^h \subseteq S'$, where $S'$ is the intersection of $S^P$ with boxes that are hit by $S^U$ (see the description of the algorithm $\mathcal{A}$). Therefore, since $S^h$ is an $\varepsilon'$-net for $\mathcal{C}\Delta\mathcal{C}$ with respect to $\mathcal{D}_{+,h}$, then so is $S'$. Hence, with probability at least $1 - 3\delta' = 1 - \delta/2$ the set $S'$ is an $\varepsilon'$-net for $\mathcal{C}\Delta\mathcal{C}$ with respect to $\mathcal{D}_{+,h}$. Now note that an $\varepsilon'$-net for $\mathcal{C}\Delta\mathcal{C}$ with respect to $\mathcal{D}_{+,h}$ is an $2\varepsilon'$-net with respect to $\mathcal{D}_+$ as every set of $\mathcal{D}_+$-weight at least $2\varepsilon'$ has $\mathcal{D}_{+,h}$ weight at least $\varepsilon'$, by definition of $\mathcal{X}^h$ and $\mathcal{D}_{+,h}$.

Next, let $c^\star \in \mathcal{C}$ denote the $\gamma$-margin classifier with $\mathrm{err}_{\mathcal{D}}(c^\star) = 0$. We show that $S^P \subseteq c^\star$. Note that every box in $\mathcal{B}$ is labeled homogeneously with label 1 or label 0 by the labeling function $l$ as $l$ is a $\gamma$-margin classifier as well. Let $s \in S'$ be a sample point and $A_s \in \mathcal{B}$ be the box that contains $s$. As $c^\star$ is a $\gamma$-margin classifier and $\mathcal{D}_{\mathcal{X}}\left(A_s\right) > 0$ ($A_s$ was hit by $S^U$ by the definition of $S'$), $A_s$ is labeled homogeneously by $c^\star$ as well and as $\mathrm{err}_{\mathcal{D}}(c^\star) = 0$ this label has to correspond to the labeling by $l$. Thus $c^\star(s) = l(s) = 1$ for all $s \in S'$. This means that $S' \subseteq c^\star$. According to Lemma 5 for some $M > 1$ since $S' \subseteq c^\star$ and $a \geq M\left(\frac{d\ln(1/\varepsilon) + \ln(1/\delta)}{\varepsilon}\right)$ as long as $S'$ is $\varepsilon/14$-net for $\mathcal{C}\Delta\mathcal{C}$ w.r.t. $\mathcal{D}_+$, with probability $1 - \delta/2$ we have $\mathrm{err}_{\mathcal{D}}(c) \leq \varepsilon$. This completes the proof.

**Proof of Claim 1:** Let $A$ be a heavy box, thus $\mathcal{D}_+(A) \geq \varepsilon'/|\mathcal{B}|$, therefore $\mathcal{D}_{\mathcal{X}}(A) \geq \pi \cdot \varepsilon'/|\mathcal{B}|$. Then, when drawing an i.i.d. sample $S^U$ from $\mathcal{D}_{\mathcal{X}}$, the probability of not hitting $A$ is at most $\left(1 - (\pi\varepsilon'/|\mathcal{B}|)\right)^a$. Now the union bound implies that the probability that there is a box in $\mathcal{B}^h$ that does not get hit by $X_U$ is bounded by

$$\left|\mathcal{B}^h\right|\left(1 - (\pi \cdot \varepsilon'/|\mathcal{B}|)\right)^a \leq |\mathcal{B}|\left(1 - (\pi \cdot \varepsilon'/|\mathcal{B}|)\right)^a$$
$$\leq |\mathcal{B}|e^{-\pi \cdot \varepsilon' \cdot a/|\mathcal{B}|}$$

Thus, if $a \geq \frac{|\mathcal{B}|\ln\left(|\mathcal{B}|/\delta'\right)}{\pi \cdot \varepsilon'} = \frac{28(\sqrt{k}/\gamma)^k \ln\left(6(\sqrt{k}/\gamma)^k/\delta\right)}{\pi\varepsilon}$, then $S^U$ will hit every heavy box with probability at least $1 - \delta'$.

**Proof of Claim 2:** Let $S^h := S^P \mid \mathcal{X}^h$. Note that, as $S^P$ is an i.i.d. $\mathcal{P}$ sample, we can consider $S^h$ to be an i.i.d. $\mathcal{P}_h$ sample. We have the following bound on the weight ratio between $\mathcal{P}_h$ and $\mathcal{D}_{+,h}$:

$$R_{\mathcal{I}}\left(\mathcal{P}_h, \mathcal{D}_{+,h}\right) = \inf_{A \in \mathcal{I}, \mathcal{D}_{+,h}(A) > 0} \frac{\mathcal{P}_h(A)}{\mathcal{D}_{+,h}(A)}$$
$$= \inf_{A \in \mathcal{I}, \mathcal{D}_{+,h}(A) > 0} \frac{\mathcal{P}(A)}{\mathcal{D}_+(A)} \frac{\mathcal{D}_+\left(\mathcal{X}^h\right)}{\mathcal{P}\left(\mathcal{X}^h\right)}$$
$$\geq r \frac{\mathcal{D}_+\left(\mathcal{X}^h\right)}{\mathcal{P}\left(\mathcal{X}^h\right)} \geq r\left(1 - \varepsilon'\right)$$

where the last inequality holds as $\mathcal{D}_+\left(\mathcal{X}^h\right) \geq \left(1 - \varepsilon'\right)$ and $\mathcal{P}\left(\mathcal{X}^h\right) \leq 1$. Note that every element in $\mathcal{C}\Delta\mathcal{C}$ can be partitioned into elements from $\mathcal{I}$, therefore we obtain the same bound on the weight ratio for the symmetric differences of $\mathcal{C}$: $R_{\mathcal{C}\Delta\mathcal{C}}\left(\mathcal{D}_{+,h}, \mathcal{D}_{+,h}\right) \geq r\left(1 - \varepsilon'\right)$.

As we argued in Corollary 6, it is well known that there is a constant $M > 1$ such that, conditioned on $S^h$ having size at least $n' := M\left(\frac{d\ln\left(1/\left(r(1-\varepsilon')\varepsilon'\right)\right) + \ln\left(1/\delta'\right)}{r(1-\varepsilon')\varepsilon'}\right)$, with probability at least $1 - \delta'$ it is a $r\left(1 - \varepsilon'\right)\varepsilon'$-net with respect to $\mathcal{P}_h$ and thus an $\varepsilon'$-net with respect to $\mathcal{D}_{+,h}$ by Lemma 7.

Thus, it remains to show that with probability at least $1 - \delta'$ we have $\left|S^h\right| \geq n'$. As we have $\mathcal{P}\left(\mathcal{X}^h\right) \geq r\left(1 - \varepsilon'\right)$, we can view the sampling of the points of $S^P$ and checking whether they hit $\mathcal{X}^h$ as a Bernoulli variable with mean $\mu = \mathcal{P}\left(\mathcal{X}^h\right) \geq r\left(1 - \varepsilon'\right)$. Thus, by Hoeffding's inequality (see Theorem 22) we have that for all $t > 0$, $\Pr\left(\mu|S^P| - \left|S^h\right| \geq t|S^P|\right) \leq \mathrm{e}^{-2t^2|S^P|}$. If we set $r' = r\left(1 - \varepsilon'\right)$, assume $|S^P| \geq \frac{2n'}{r'}$ and set $t = r'/2$, we obtain $\Pr\left(\left|S^h\right| < n'\right) \leq \Pr\left(\mu|S^P| - \left|S^h\right| \geq \frac{r'}{2}|S^P|\right) \leq \mathrm{e}^{-\frac{r'^2|S^P|}{2}}$.

Now $|S^P| \geq \frac{2n'}{r'} > \frac{2\left(d\ln\left(1/\left(r(1-\varepsilon')\varepsilon'\right)\right)+\ln\left(1/\delta'\right)\right)}{r^2(1-\varepsilon')^2\varepsilon'}$ implies that $\mathrm{e}^{-\frac{r'^2|S^P|}{2}} \leq \delta'$, thus we have shown that $S^h$ is an $\varepsilon'$-net of $\mathcal{C}\Delta\mathcal{C}$ with probability at least $\left(1 - \delta'\right)^2 \geq 1 - 2\delta'$. $\qquad\square$

# D  Missing Proofs from Section 5

**Theorem 13.** *Let $\mathcal{C}$ be a concept class with $\mathrm{VCD}(\mathcal{C}) = d$ where $d \geq 4$. Consider $\mathcal{W}$ to be the set of duos $(\mathcal{D}, \mathcal{D}_+)$ with $\mathcal{D}[\{(x, 1) : x \in \mathcal{X}\}] = 0.5$. Then $\mathcal{C}$ is PU learnable over $\mathcal{W}$ with sample complexity $m_{\mathcal{C}}^{unlabel}(\varepsilon, \delta), m_{\mathcal{C}}^{pos}(\varepsilon, \delta) = \Omega\left(\frac{d+\ln(1/\delta)}{\varepsilon^2}\right)$ and $m_{\mathcal{C}}^{unlabel}(\varepsilon, \delta), m_{\mathcal{C}}^{pos}(\varepsilon, \delta) = O\left(\frac{d\ln(1/\varepsilon)+\ln(1/\delta)}{\varepsilon^2}\right)$.*

We prove the theorem by reducing it to a problem we refer to as the *generalized weighted die problem*. This approach is inspired by Theorem 1 of Ben-David and Ben-David (2011), in which a learning problem –referred to as learning a classifier when a labeling is known (KLCL)– is reduced to the *weighted die problem*.

In the weighted die problem, a die has one face that is slightly biased, and the goal is to identify the biased face using $m$ rolls. In the generalized weighted die problem, multiple faces of the die may be slightly biased, and the goal is to identify all of them.

We now formally define the generalized weighted die problem. Before doing so, we introduce some necessary notation. For any $k \in \mathbb{N}$, define the set $V_k := 2^{[k]} \setminus \{[k], \varnothing\}$. Next, define two weighting functions $w^-, w^+ : V_k \to [0, 1]$ as follows: for any $O \in V_k$ with $|O| \leq \frac{k}{2}$, set

$$w^-(O) := 1 \quad \text{and} \quad w^+(O) := \frac{|O|}{k - |O|},$$

and for $|O| > \frac{k}{2}$, set

$$w^-(O) := \frac{k - |O|}{|O|} \quad \text{and} \quad w^+(O) := 1.$$

**Definition 7** (Generalized Weighted Die Problem). *We define the generalized weighted die problem with parameters $k \geq 2$ and $\varepsilon \in (0, 1)$ as follows. For each $O \in V_k$ suppose there is a die with $k$ faces, with probability of each face $j \in [k]$ being*

$$P_O(j) = \begin{cases} \frac{1 - w^-(O)\varepsilon}{k} & j \in O \\ \frac{1 + w^+(O)\varepsilon}{k} & j \notin O \end{cases}$$

*The die $O$ is rolled $m$ times. A learner gets the outcome of these $m$ die rolls, and its target is to output a $h \in \{0, 1\}^k$ which minimizes $err(h) := \frac{1}{k}\left(\sum_{j \in O} h_j w^-(O) + \sum_{j \notin O}(1 - h_j)w^+(O)\right)$.*

**Theorem 31.** *Suppose the die $O$ in the generalized weighted die problem with parameters $k \geq 32$ and $\varepsilon \in (0, 1)$ is drawn uniformly at random from the set $V_k$. If the number of rolls is at most $k\left\lfloor\frac{1}{2\varepsilon^2}\right\rfloor$, then any algorithm for the generalized weighted die problem incurs an error of at least $\frac{1}{160}$ with probability greater than $\frac{1}{320}$.*

*Proof.* Set $m = k\left\lfloor\frac{1}{2\varepsilon^2}\right\rfloor$. Note that it is multiplied by $k$. With an argument similar to Lemma 5.1 of Anthony and Bartlett (2009) and Theorem 2 of Ben-David and Ben-David (2011), it is easy to see that the perfect learner $L_0$ predicts $j$ to have positive bias iff the number of samples from $j$ is bigger than $m/k$.

For a sample roll drawn from $P_O^m$ define $c_S$ to be the output of $L_0$. Define event $E$ to be the event that $\min(|O|, k - |O|) \geq \frac{k}{4}$

$$\mathbb{E}_{O \sim U_{V_k}} \left[ \mathbb{E}_{S \sim P_O^m} \left[ err(c_S) \right] \right]$$
$$= \mathbb{E}_{O \sim U_{V_k}} \left[ \mathbb{E}_{S \sim P_O^m} \left[ err(c_S) \right] \mid E \right] \Pr_{O \sim U_{V_k}} [E] \tag{24}$$

Note that since two set in $2^{[k]} \setminus V_k$ are not in event $E$ it is clear that $\Pr_{O \sim U_{V_k}} [E] \geq Pr_{O \sim U_{2^{[k]}}} [E]$. Moreover, note that when $O$ is chosen uniformly from $U_{2^{[k]}}$, $|O|$ can be looked upon as a random variable drawn from $Bin(k, 1/2)$. Therefore, as long as $k \geq 32$, using Multiplicative Chernoff bound (Lemma 21) we can derive

$$\Pr_{O \sim U_{2^{[k]}}} [E] \geq \left( 1 - 2 \exp\left( -\frac{k}{16} \right) \right) > \frac{1}{2} \tag{25}$$

Then we try to bound $\mathbb{E}_{O \sim U_{V_k}} \left[ \mathbb{E}_{S \sim P_O^m} \left[ err(c_S) \right] \mid E \right]$. For every $i \in [k]$ denote $s_i$ to be the number of $i$ in $S$. Then fix any $O \in V_k$ such that $\min(|O|, k - |O|) \geq \frac{k}{4}$, and any $i \in O$. Note that $L_0$ will make a wrong prediction for the face $i$ iff $s_i \geq m/k$. Since $s_i$ is $Bin(p, m)$ where $p = \frac{1 - w^-(O)\varepsilon}{k}$, using Slud's inequality (Lemma 24) we can derive

$$\Pr[s_i \geq m/k] \geq P \left[ Z \geq \frac{m\varepsilon w^-(O)}{\sqrt{m(1+\varepsilon)(k-1+\varepsilon)}} \right]$$
$$\overset{(i)}{\geq} P \left[ Z \geq \frac{m\varepsilon}{\sqrt{m(k-1)}} \right]$$
$$\geq P \left[ Z \geq \sqrt{\frac{2m\varepsilon^2}{k}} \right] \tag{26}$$
$$\overset{(ii)}{\geq} \frac{1}{2} \left( 1 - \sqrt{1 - \exp\left( -\frac{2m\varepsilon^2}{k} \right)} \right)$$
$$\geq \frac{1}{2} \left( 1 - \sqrt{1 - e^{-1}} \right) > 0.1$$

Where $Z \sim N(0, 1)$ is a normally distributed random variable with mean of 0 and standard deviation of 1. Moreover, (i) is due to $w^-(O) \leq 1$ and $\varepsilon \geq 0$, and (ii) is due to Lemma 25. Thus,

$$\mathbb{E}_{S \sim P_O^m} \left[ err(c_S) \right] \geq \sum_{i \in O} \frac{\Pr[s_i \geq m/k]}{k}$$
$$> |O| \cdot w^-(O) \cdot \frac{0.1}{k} \tag{27}$$
$$= \min(|O|, k - |O|) \cdot \frac{0.1}{k}$$
$$\geq \frac{1}{40}$$

Combining this with (24), (25), we conclude that $\mathbb{E}_{O \sim U_{V_k}} [\mathbb{E}_{S \sim P_O^m} \left[ err(c_S) \right]] > \frac{1}{80}$. Using Lemma 20 since $err(c_S)$ is always less than 2, we have

$$Pr_{O \sim U_{V_k}, S \sim P_m^O} \left[ err(c_S) \geq \frac{1}{160} \right] > \frac{1}{320}$$

which completes the proof. □

Consider the specific case where $k = 2$. Then $V_2 = \{\{1\}, \{2\}\}$. In this case, one of the faces always has probability $p = \frac{1+\varepsilon}{2}$, and the other has probability $p = \frac{1-\varepsilon}{2}$. Any algorithm with error less than $\frac{1}{2}$ must correctly identify which face has the higher probability. In other words, in this special case, the generalized weighted die problem reduces to Lemma 5.1 of Anthony and Bartlett (2009), stated below.

**Lemma 32** (Lemma 5.1 of Anthony and Bartlett (2009)). *Let $\varepsilon < \frac{1}{2}$, $\delta < \frac{1}{4}$. Suppose $y = U_{\{-1,+1\}}$. Then if $m < \frac{1}{4\varepsilon^2} \ln\left(\frac{1}{4\delta}\right)$ there is no algorithm that can predict $y$ with probability more than $1 - \delta$ using a sample $S \sim Bin(m, \alpha)$ where $\alpha = \frac{1+y\varepsilon}{2}$.*

**Corollary 33.** *Let $\varepsilon < \frac{1}{2}$, $\delta < \frac{1}{4}$. Suppose the die $O$ in the generalized weighted die problem with parameters $k = 2$ and $\varepsilon$ is drawn uniformly at random from the set $V_2$. If the number of rolls is at most $\frac{1}{4\varepsilon^2} \ln\left(\frac{1}{4\delta}\right)$, then any algorithm for the generalized weighted die problem incurs an error of at least $\frac{1}{2}$ with probability greater than $\delta$.*

*Proof of Theorem 13.* The upper bounds for Theorem 13 are already known Du Plessis et al. (2015). We only focus on the lower-bounds.

Consider any $k \in \mathbb{N}$, and consider any $B \subseteq \mathcal{X}$ of size $2k$, and any $\rho \in (0,1)$. We randomly divide $B$ into two halves of size $k$ named $B^1 = \{x_1^1, x_2^1, ..., x_k^1\}$ and $B_2 = \{x_1^1, x_2^2, ..., x_k^2\}$. Let $\rho \in (0,1)$, and for any $O^1, O^2 \in V_k$, we define distribution $\mathcal{D}_{O^1, O^2}$ over $\mathcal{X} \times \{0,1\}$ for all $(x,y) \in \mathcal{X} \times \{0,1\}$ as

$$\mathcal{D}_{O^1,O^2}(x,y) = \begin{cases} \frac{1}{4k} & \text{if } x \in B_1, \text{ and } y = 1 \\ \frac{1+w^+(O^1)\rho}{4k} & \text{if } x = x_i^1 \text{ and } i \notin O^1 \text{ and } y = 0 \\ \frac{1-w^-(O^1)\rho}{4k} & \text{if } x = x_i^1 \text{ and } i \in O^1 \text{ and } y = 0 \\ \frac{1+\frac{(2y-1)w^+(O^2)\rho}{2}}{4k} & \text{if } x = x_i^2 \text{ and } i \notin O^2 \\ \frac{1-\frac{(2y-1)w^-(O^2)\rho}{2}}{4k} & \text{if } x = x_i^2 \text{ and } i \in O^2 \\ 0 & \text{o.w.} \end{cases}$$

We define $\mathcal{W}_{\rho,B}^{agno} := \{(\mathcal{D}_{O^1,O^2}, \mathcal{D}_{+,O^1,O^2}) \mid O^1, O^2 \in V_k\}$. Notice that every $\mathcal{D}_{O^1,O^2}$ satisfies $\mathcal{D}_{O^1,O^2}[\{(x,1) : x \in \mathcal{X}\}] = \frac{1}{2}$. Moreover, it is easy to see that the error with respect to $\mathcal{D}_{O^1,O^2}$ is minimized by any function $f : \mathcal{X} \to \{0,1\}$ that satisfies $f \cap B^1 = O^1$ and $f \cap B^2 = B^2 \setminus O^2$. Therefore, for any function $c$ we have

$$\text{err}_{\mathcal{D}_{O^1,O^2}}(c) - \min_{\text{functions } f} \text{err}_{\mathcal{D}_{O^1,O^2}}(f) = \frac{\rho}{4k}\left( \sum_{i \in O^1} \mathbb{1}[c(x_i^1) \neq 1]w^-(O^1) + \sum_{i \notin O^1} \mathbb{1}[c(x_i^1) \neq 0]w^+(O^1) \right.$$

$$\left. + \sum_{i \in O^2} \mathbb{1}[c(x_i^2) \neq 0]w^-(O^2) + \sum_{i \notin O^2} \mathbb{1}[c(x_i^2) \neq 1]w^+(O^2) \right)$$
(28)

The following lemma reduces both the number of unlabeled and positive samples in PU learning over $\mathcal{W}_{\rho,B}^{agno}$ to the generalized weighted die problem.

**Lemma 34.** *Let $\varepsilon, \delta, \rho \in (0,1)$ and $m, n \in \mathbb{N}$. Suppose there exists a PU learner $\mathcal{A}$ such that for all $(\mathcal{D}, \mathcal{D}_+) \in \mathcal{W}_{\rho,B}^{agno}$, $b \geq m$, $a \geq n$ satisfies*

$$Pr_{S^P \sim \mathcal{D}_+^b, S^U \sim \mathcal{D}_\mathcal{X}^a}[\text{err}_\mathcal{D}(\mathcal{A}(S^P, S^U)) - \min_{\text{functions } f} \text{err}_\mathcal{D}(f) \geq \varepsilon] \leq \delta$$

*Then, (i) there exists a learner that with probability $1 - \delta$ achieves error less than $\frac{4\varepsilon}{\rho}$ using $m$ rolls for a weighted generalized with parameters $k$ and $\rho$; (ii) there exists a learner that with probability $1 - \delta$ achieves error less than $\frac{4\varepsilon}{\rho}$ using $n$ rolls for a weighted generalized with parameters $k$ and $\frac{\rho}{2}$.*

*Proof.* We prove the first part, and the second part is identical to the first part. Fix any die corresponding to $O \in V_k$. Suppose $S = \{i_1, i_2, ..., i_m\}$ is a roll of size $m$ from $P_O$. Let $O' = \{1\}$. First notice that for any $x \in \mathcal{X}$ we have

$$\mathcal{D}_{+,O,O'} := \begin{cases} \frac{1}{2k} & x \in B^1 \\ \frac{1+\frac{\rho}{2(k-1)}}{2k} & \text{if } x = x_i^2 \text{ and } i \neq 1 \\ \frac{1-\frac{\rho}{2}}{2k} & \text{if } x = x_1^2 \\ 0 & \text{o.w.} \end{cases}$$

Moreover,

$$\mathcal{D}_{\mathcal{X},O,O'}(x \mid x \in B^2) = \begin{cases} \frac{1+\frac{\rho}{k-1}}{k} & \text{if } x = x_i^2 \text{ and } i \neq 1 \\ \frac{1-\rho}{k} & \text{if } x = x_1^2 \\ 0 & \text{o.w.} \end{cases} \tag{29}$$

Notice that both $\mathcal{D}_{+,O,O'}$ and $\mathcal{D}_{\mathcal{X},O,O'}(.\mid x \in B^2)$ are independent from $O$, and thus a learner can collect samples from them without requiring knowledge about $O$. Next, we define $S^P$ to be an i.i.d sample of size $n$ from $\mathcal{D}_{+,O,O'}$. Then, construct $S^U$ as follows. Initialize $j = 0$. At each step, flip an unbiased coin. If it lands heads, increment $j$ by 1 and add $x_{i_j}^1$ to $S^U$. If it lands tails, draw a sample from $\mathcal{D}_{\mathcal{X},O,O'}(\cdot \mid x \in B^2)$ and add it to $S^U$. Repeat this process until $j = m$.

Moreover, observe that $\mathcal{D}_{\mathcal{X},O,O'}(\cdot \mid x \in B^1) = P_O$. Hence, each $x_{i_j}^1$ is an independent sample from $\mathcal{D}_{\mathcal{X},O,O'}(\cdot \mid x \in B^1)$. Since $\mathcal{D}_{\mathcal{X},O,O'}(B^1) = \frac{1}{2}$, it follows that $S^U$ is an i.i.d. sample from $\mathcal{D}_{\mathcal{X},O,O'}$. Note that due to lemma's assumption, since $|S^U| \geq m$, $|S^P| \geq n$ with probability $1 - \delta$ we have

$$\text{err}_{\mathcal{D}_{O,O'}}(\mathcal{A}(S^P, S^U)) - \min_{\text{functions } f} \text{err}_{\mathcal{D}_{O,O'}}(f) < \varepsilon$$

We define our leaner $h \in \{0,1\}^k$ as $h_i := (1 - \mathcal{A}(S^P, S^U)(x_i^1))$ for $i \in [k]$. Due to (28) we have

$$\text{err}_{\mathcal{D}_{O,O'}}\left(\mathcal{A}(S^P, S^U)\right) - \min_{\text{functions } f} \text{err}_{\mathcal{D}_{O,O'}}(f)$$

$$\geq \frac{\rho}{4k}\left(\sum_{i \in O} \mathbb{1}\left[\mathcal{A}(S^P, S^U)(x_i^1) \neq 1\right] w^-(O) + \sum_{i \notin O} \mathbb{1}\left[\mathcal{A}(S^P, S^U)(x_i^1) \neq 0\right] w^+(O)\right)$$

$$= \frac{\rho}{4}\text{err}(h)$$

Thus $\text{err}(h) \leq \frac{4\varepsilon}{\rho}$ with probability $1 - \delta$.

$\square$

Then, we first show that for $d \geq 64$ we have $m_{\mathcal{C}}^{unlabel}(\varepsilon, \frac{1}{320}) \geq \lfloor\frac{d}{2}\rfloor\lfloor\frac{1}{819200\varepsilon^2}\rfloor, m_{\mathcal{C}}^{pos}(\varepsilon, \frac{1}{320}) \geq \lfloor\frac{d}{2}\rfloor\lfloor\frac{1}{204800\varepsilon^2}\rfloor$. Let $\varepsilon < \frac{1}{640}$, and set $k = \lfloor\frac{d}{2}\rfloor$. Let $B$ be any subset of $\mathcal{X}$ of size $2k$ that is shattered by $\mathcal{C}$. Define $\rho = 640\varepsilon$, and let the number of unlabeled examples be $a = k\lfloor\frac{1}{2\rho^2}\rfloor$, while the number of positive examples is any $b \in \mathbb{N}$. For the sake of contradiction, suppose there exists a PU learner $\mathcal{A}$ such that for all $(\mathcal{D}, \mathcal{D}_+) \in \mathcal{W}_{\rho,B}^{agno}$ satisfies

$$Pr_{S^P \sim \mathcal{D}_+^b, S^U \sim \mathcal{D}_{\mathcal{X}}^a}[\text{err}_{\mathcal{D}}(\mathcal{A}(S^P, S^U)) - \min_{c \in \mathcal{C}} \text{err}_{\mathcal{D}}(\mathcal{A}(S^P, S^U)) \geq \varepsilon] \leq \frac{1}{320}$$

Using Lemma 34 generalized weighted die problem with parameters $k$ and $\rho$ can achieve error $\frac{1}{160}$ with probability $\frac{1}{320}$ with $a$ rolls. Assuming $k \geq 32$ (which holds as long as $d \geq 64$) this contradicts Theorem 13. Therefore, we can conclude that no matter how many positive samples the learner receives, the sample complexity of unlabeled examples should be at least $m_{\mathcal{C}}^{unlabel}(\varepsilon, \frac{1}{320}) \geq \lfloor\frac{d}{2}\rfloor\lfloor\frac{1}{819200\varepsilon^2}\rfloor$. Similarly, we can see that no matter how many unlabeled samples the learner receives, as long as $d \geq 64$, the sample complexity of unlabeled examples should be at least $m_{\mathcal{C}}^{pos}(\varepsilon, \frac{1}{320}) \geq \lfloor\frac{d}{2}\rfloor\lfloor\frac{1}{204800\varepsilon^2}\rfloor$.

Finally, we show that for $d \geq 4$, we have $m_{\mathcal{C}}^{unlabel}(\varepsilon, \delta) \geq \frac{1}{256\varepsilon^2}\ln\left(\frac{1}{4\delta}\right), m_{\mathcal{C}}^{pos}(\varepsilon, \delta) \geq \frac{1}{64\varepsilon^2}\ln\left(\frac{1}{4\delta}\right)$. For $\varepsilon < \frac{1}{16}$ and $\delta < \frac{1}{4}$, let $k = 2$ and let $B$ be any subset of size 4 shattered by $\mathcal{C}$. Define $\rho = 8\varepsilon$, and let the number of unlabeled examples be $a = \frac{1}{4\rho^2}\ln\left(\frac{1}{4\delta}\right)$, while the number of positive examples is any $b \in \mathbb{N}$. For the sake of contradiction, suppose there exists a PU learner $\mathcal{A}$ such that for all $(\mathcal{D}, \mathcal{D}_+) \in \mathcal{W}_{\rho,B}^{agno}$ satisfies

$$Pr_{S^P \sim \mathcal{D}_+^b, S^U \sim \mathcal{D}_{\mathcal{X}}^a}[\text{err}_{\mathcal{D}}(\mathcal{A}(S^P, S^U)) - \min_{c \in \mathcal{C}} \text{err}_{\mathcal{D}}(\mathcal{A}(S^P, S^U)) \geq \varepsilon] \leq \delta$$

Then, by Lemma 34, the generalized weighted die problem with parameters 2 and $\rho$ would achieve error less than $\frac{1}{2}$. Since $\rho < \frac{1}{2}$ and $\delta < \frac{1}{4}$, this contradicts Corollary 32. Therefore, we can conclude

that no matter how many positive samples the learner receives, the sample complexity of unlabeled examples should be at least $m_{\mathcal{C}}^{unlabel}(\varepsilon,\delta) \geq \frac{1}{256\varepsilon^2}\ln\left(\frac{1}{4\delta}\right)$. Similarly, we can see that no matter how many unlabeled samples the learner receives, as long as $d \geq 4$ the sample complexity of unlabeled examples should be at least $m_{\mathcal{C}}^{pos}(\varepsilon,\delta) \geq \frac{1}{64\varepsilon^2}\ln\left(\frac{1}{4\delta}\right)$.

**Theorem 16.** *Let $\mathcal{C}$ be any concept class over domain $\mathcal{X}$ with VC dimension d, and let $\mathcal{P} = \mathcal{D}_+$. Given any $\gamma \geq \alpha$, denote $c^{PU} = \operatorname{argmin}_{c \in \mathcal{C}} \hat{\text{err}}^\gamma(c)$. There exists $M > 1$ such that for all $c \in \mathcal{C}$, if $|S^P|, |S^U| > \frac{M(d+\ln(1/\delta))}{\varepsilon^2}$, then with probability $1 - 4\delta$ we have*

$$\text{err}_{\mathcal{D}}(c^{PU}) \leq \max\left(\frac{\gamma - \alpha}{\alpha}, \frac{\alpha}{\gamma - \alpha}\right)(\text{err}_{\mathcal{D}}(c) + 2(1 + \gamma)\varepsilon)$$

*Proof.* Using standard PAC theory, for all $c' \in \mathcal{C}$ there exists some $M > 1$ such that if $|S^U| > \frac{M(d\ln(1/\varepsilon) + \ln(1/\delta))}{\varepsilon^2}$, then with probability $1 - \delta$ we have $\left|\frac{|c'|S^U|}{a} - Pr(c'(x) = 1)\right| \leq \varepsilon$ (e.g. see Shalev-Shwartz and Ben-David (2014)). Similarly, there exists some $M > 1$ such that if $|S^P| > \frac{M(d\ln(1/\varepsilon) + \ln(1/\delta))}{\varepsilon^2}$, then with probability $1 - \delta$ we have $\left|\text{err}_{\mathcal{D}}^+(c') - \frac{b - |c'|S^P|}{b}\right| \leq \varepsilon$. Combining these two facts, with probability $1 - 2\delta$ for all $c' \in \mathcal{C}$ we derive

$$|Pr(c'(x) = 1) + \gamma\,\text{err}_{\mathcal{D}}^+(c') - \hat{\text{err}}^\gamma(c')| \leq (1 + \gamma)\varepsilon \tag{30}$$

Then, for any $c' \in \mathcal{C}$ we have

$$\Pr(c'(x) = 1) = \Pr(y = 1) - \Pr(y = 1, c'(x) = 0) + \Pr(c'(x) = 1, y = 0)$$
$$= \alpha - \alpha\,\text{err}_{\mathcal{D}}^+(c') + (1 - \alpha)\,\text{err}_{\mathcal{D}}^-(c') \tag{31}$$

Now, fix any $c \in \mathcal{C}$. For the cases where $\gamma \geq 2\alpha$, with probability $1 - 2\delta$ we have

$$\begin{aligned}
\alpha + \text{err}_{\mathcal{D}}(c^{PU}) &= \alpha + \alpha\,\text{err}_{\mathcal{D}}^+(c^{PU}) + (1 - \alpha)\,\text{err}_{\mathcal{D}}^-(c^{PU}) \\
&\leq \alpha + (\gamma - \alpha)\,\text{err}_{\mathcal{D}}^+(c^{PU}) + (1 - \alpha)\,\text{err}_{\mathcal{D}}^-(c^{PU}) \\
&\stackrel{(31)}{=} Pr(c^{PU}(x) = 1) + \gamma\,\text{err}_{\mathcal{D}}^+(c^{PU}) \\
&\stackrel{(30)}{\leq} \hat{\text{err}}^\gamma(c^{PU}) + (1 + \gamma)\varepsilon \\
&\stackrel{(i)}{\leq} \hat{\text{err}}^\gamma(c) + (1 + \gamma)\varepsilon \\
&\stackrel{(30)}{\leq} Pr(c(x) = 1) + \gamma\,\text{err}_{\mathcal{D}}^+(c) + 2(1 + \gamma)\varepsilon \\
&\stackrel{(31)}{=} \alpha + (\gamma - \alpha)\,\text{err}_{\mathcal{D}}^+(c) + (1 - \alpha)\,\text{err}_{\mathcal{D}}^-(c) + 2(1 + \gamma)\varepsilon \\
&= \alpha + (\gamma - 2\alpha)\,\text{err}_{\mathcal{D}}^+(c) + \text{err}_{\mathcal{D}}(c) + 2(1 + \gamma)\varepsilon \\
&\stackrel{(ii)}{\leq} \alpha + \left(1 + \frac{\gamma - 2\alpha}{\alpha}\right)\text{err}_{\mathcal{D}}(c) + 2(1 + \gamma)\varepsilon.
\end{aligned} \tag{32}$$

Where (i) is due to the definition of $c^{PU}$ and (ii) is due to the fact that $\text{err}_{\mathcal{D}}^+(c) \leq \frac{\text{err}_{\mathcal{D}}(c)}{\alpha}$. For the cases where $\gamma < 2\alpha$ with probability $1 - 2\delta$ we have

$$\begin{aligned}
\alpha + \frac{\gamma - \alpha}{\alpha}\,\text{err}_{\mathcal{D}}(c^{PU}) &= \alpha + (\gamma - \alpha)\,\text{err}_{\mathcal{D}}^+(c^{PU}) + (1 - \alpha)\frac{\gamma - \alpha}{\alpha}\,\text{err}_{\mathcal{D}}^-(c^{PU}) \\
&\leq \alpha + (\gamma - \alpha)\,\text{err}_{\mathcal{D}}^+(c^{PU}) + (1 - \alpha)\,\text{err}_{\mathcal{D}}^-(c^{PU}) \\
&\stackrel{(i)}{\leq} \alpha + (\gamma - \alpha)\,\text{err}_{\mathcal{D}}^+(c) + (1 - \alpha)\,\text{err}_{\mathcal{D}}^-(c) + 2(1 + \gamma)\varepsilon \\
&\leq \alpha + \text{err}_{\mathcal{D}}(c) + 2(1 + \gamma)\varepsilon
\end{aligned} \tag{33}$$

where (i) is derived similarly to (32). This completes the proof. $\qquad\square$

**Theorem 35** (Ben-David et al. (2010)). *Let $\mathcal{C}$ be a VC-dimension d concept class over domain $\mathcal{X}$, and let $\mathcal{Q}_S$ and $\mathcal{Q}_T$ be distributions over $\mathcal{X} \times \{0, 1\}$. Then, with probability at least $1 - \delta$, for every $c \in \mathcal{C}$:*

$$\text{err}_{\mathcal{Q}_S}(c) \leq \text{err}_{\mathcal{Q}_T}(c) + d_{\mathcal{C}\triangle\mathcal{C}}(\mathcal{Q}_{\mathcal{X},S}, \mathcal{Q}_{\mathcal{X},T}) + \lambda$$

*where $\lambda := \inf_{c \in \mathcal{C}} \text{err}_{\mathcal{Q}_S}(c) + \text{err}_{\mathcal{Q}_T}(c)$.*

**Theorem 19.** *Let $\mathcal{C}$ be any concept class over domain $\mathcal{X}$ with VC dimension $d$, and let $\mathcal{P}$ be any arbitrary distribution. Given any $\gamma \geq \alpha$, denote $c^{PU} = \mathrm{argmin}_{c \in \mathcal{C}} \, \hat{\mathrm{err}}^{\gamma}(c)$. There exists $M > 1$ such that for all $c \in \mathcal{C}$, if $|S^P|, |S^U| > \frac{M(d + \ln(1/\delta))}{\varepsilon^2}$, then with probability $1 - 4\delta$ we have*

$$\mathrm{err}_{\mathcal{D}}(c^{PU}) \leq \max\left(\frac{\gamma - \alpha}{\alpha}, \frac{\alpha}{\gamma - \alpha}\right)\left(err_{\mathcal{D}}(c) + 2(1 + \gamma)\varepsilon + 2\gamma\left(\lambda^P + d_{\mathcal{C}\triangle\mathcal{C}}(\mathcal{P}, \mathcal{D}_+)\right)\right)$$

*Where $\lambda^P := \min_{c \in \mathcal{C}}\left(\mathrm{err}_{\mathcal{D}}^+(c) + \mathrm{err}_{\mathcal{P}}(c, 1)\right)$ and $\mathrm{err}_{\mathcal{P}}(c, 1) := \mathrm{Pr}_{x \sim \mathcal{P}}(c(x) \neq 1)$.*

*Proof.* Note that similar to (30), again we know there exists some $M > 1$ such that for all $c' \in C$ with probability $1 - 2\delta$

$$|\mathrm{Pr}(c'(x) = 1) + \gamma \, \mathrm{err}_{\mathcal{P}}(c', 1) - \hat{\mathrm{err}}^{\gamma}(c')| \leq (1 + \gamma)\varepsilon \tag{34}$$

Fix any $c \in \mathcal{C}$. Thus, for the cases where $\gamma \geq 2\alpha$ similar to (32) with probability $1 - 2\delta$ we have

$$
\begin{aligned}
\alpha + \mathrm{err}_{\mathcal{D}}(c^{PU}) &\overset{(i)}{\leq} \mathrm{Pr}(c^{PU}(x) = 1) + \gamma \, \mathrm{err}_{\mathcal{D}}^+(c^{PU}) \\
&\overset{\text{Theorem 35}}{\leq} \mathrm{Pr}(c^{PU}(x) = 1) + \gamma \, \mathrm{err}_{\mathcal{P}}(c^{PU}, 1) + \gamma\left(\lambda^P + d_{\mathcal{C}\triangle\mathcal{C}}(\mathcal{P}, \mathcal{D}_+)\right) \\
&\overset{(34)}{\leq} \hat{\mathrm{err}}^{\gamma}(c^{PU}) + (1 + \gamma)\varepsilon + \gamma\left(\lambda^P + d_{\mathcal{C}\triangle\mathcal{C}}(\mathcal{P}, \mathcal{D}_+)\right) \\
&\overset{(31)}{\leq} \hat{\mathrm{err}}^{\gamma}(c) + (1 + \gamma)\varepsilon + \gamma\left(\lambda^P + d_{\mathcal{C}\triangle\mathcal{C}}(\mathcal{P}, \mathcal{D}_+)\right) \\
&\overset{(34)}{\leq} \mathrm{Pr}(c(x) = 1) + \gamma \, \mathrm{err}_{\mathcal{P}}(c, 1) + (1 + \gamma)\varepsilon + \gamma\left(\lambda^P + d_{\mathcal{C}\triangle\mathcal{C}}(\mathcal{P}, \mathcal{D}_+)\right) \\
&\overset{\text{Theorem 35}}{\leq} \mathrm{Pr}(c(x) = 1) + \gamma \, \mathrm{err}_{\mathcal{D}}^+(c) + (1 + \gamma)\varepsilon + 2\gamma\left(\lambda^P + d_{\mathcal{C}\triangle\mathcal{C}}(\mathcal{P}, \mathcal{D}_+)\right) \\
&\overset{(ii)}{\leq} \alpha + \left(1 + \frac{\gamma - 2\alpha}{\alpha}\right)\mathrm{err}_{\mathcal{D}}(c) + 2(1 + \gamma)\varepsilon + 2\gamma\left(\lambda^P + d_{\mathcal{C}\triangle\mathcal{C}}(\mathcal{P}, \mathcal{D}_+)\right)
\end{aligned}
\tag{35}
$$

Where (i) and (ii) are respectively due to the first two lines and last two lines of (32) of Theorem 16. Again similar to (33), in case $\gamma < 2\alpha$ with probability $1 - 2\delta$ we have

$$\alpha + \frac{\gamma - \alpha}{\alpha}\mathrm{err}_{\mathcal{D}}(c^{PU}) \leq \alpha + \mathrm{err}_{\mathcal{D}}(c) + 2(1 + \gamma)\varepsilon + 2\gamma\left(\lambda^P + d_{\mathcal{C}\triangle\mathcal{C}}(\mathcal{P}, \mathcal{D}_+)\right) \tag{36}$$

This completes the proof. $\qquad\square$

