# OpenReview forum: "Learning from positive and unlabeled examples -Finite size sample bounds"
_NeurIPS.cc/2025/Conference — NeurIPS 2025 poster_

### Official Review · Reviewer_8f7e · 2025-06-30

**Clarity:** 2
**Significance:** 3
**Originality:** 2
**Rating:** 5
**Confidence:** 3

**Summary:**

This paper provides sample complexity bounds in a general *positive unlabled* learning setting. Here the learner has access to two different distributions (from potentially known families of distributions). one generating the unlabeled data and the second generating positively labeled data. They study agnostic and realizable PAC learning variants, which previously appeared in the literature. Their main contributions are novel upper bounds when the class distribution is not known (which appears to be a standard assumption in this line of work), novel tighter lower bounds in various settings, and a novel combinatorial parameter, the *claw number*, allowing some lower bounds on the number of required unlabeled examples.

**Questions:**

Your proposition 4 strongly reminds me of learning with one-sided error [7]. There it is the same situation that learning is possible iff the intersection-closure has finite VC dimension. Is there any relationship here?

Please compare the claw number to existing combinatorial parameters beyond VC Dimension (e.g., [1,2], even invariants from convex geometry like Helly, Carathéodory, Radon numbers [3] etc). Are there any known related results? This could provide more context on the relevance of the claw number and perhaps even lead to tighter results.

The claw number based lower bounds are not fully matching the upper bounds, in particular, there is a remaining gap of $h\le d$. Do you think the upper bound can be improved from vc to claw number or vice versa? Or is the gap in general unavoidable (using these parameters)?

I think it is very interesting that you manage to get lower bound on the number of unlabeled examples. In related settings like semi-supervised [5,6] and and transductive learning [4] it is commonly known that unlabeled samples do not really in general.



[1] Bousquet, Olivier, et al. "Proper learning, Helly number, and an optimal SVM bound." COLT 2020.
[2] Hanneke, Steve. "The Star Number and Eluder Dimension: Elementary Observations About the Dimensions of Disagreement." COLT 2024.
[3] Kay, David, and Eugene W. Womble. "Axiomatic convexity theory and relationships between the Carathéodory, Helly, and Radon numbers." Pacific Journal of Mathematics 38.2 (1971): 471-485.
[4] Tolstikhin, I., & Lopez-Paz, D. (2016). Minimax lower bounds for realizable transductive classification. arXiv preprint arXiv:1602.03027.
[5] Balcan, M. F., & Blum, A. (2010). A discriminative model for semi-supervised learning. Journal of the ACM (JACM), 57(3), 1-46.
[6] Darnstädt, M., & Simon, H. U. (2011). Smart PAC-learners. Theoretical Computer Science, 412(19), 1756-1766.
[7] Natarajan, Balaubramaniam Kausik. "On learning boolean functions." Proceedings of the nineteenth annual ACM symposium on Theory of computing. 1987

**Ethical Concerns:**

["NO or VERY MINOR ethics concerns only"]

**Final Justification:**

Good paper. Authors clarified concerns.

**Limitations:**

yes

**Paper Formatting Concerns:**

.

**Quality:**

3

**Strengths And Weaknesses:**

Strengths:
* summarizes and improves various previous results on PU learning.

The main weakness I see that this paper would benefit greatly from a deeper discussion with related work and if the results would be put into a broader context. See my questions below. **When these are addressed appropriately, I am happy to raise my score.**

Some more smaller weaknesses:
* some of the proposed/improved bounds are still not tight. While the claw number is interesting, it's unclear whether it gives the optimal tight bounds. See also questions below.
* it's sometimes not fully clear when results are stated for general concept classes and when they focus on particular domains ($X=[0,1]^k$) and/or assume large margin etc.

Small comments:
* there are appear to be various smaller formatting issues. The table 1 too large (and consider using booktabs and removing the vertical lines). Some typos. Usage of dashes seems wrong.

---

> ### Author Rebuttal · Authors · 2025-07-31
>
> Thank you for your detailed feedback and insightful questions. We would like to address your questions and concerns bellow:
> - **Broader context to our contribution and extended discussion of related work**: In lines 63–70 of the paper, we briefly summarized our main high-level contributions. We will expand this summary and provide a more detailed discussion of related work in the camera-ready version.
> - **Tightness of the lower bound for unlabeled examples:**  The reviewer is right. Our lower bound for the number of unlabeled examples is not tight. A complete characterization of the sample complexity for realizable PU learning in the SCAR setup remains an open problem. That said, to the best of our knowledge, this is the first lower bound on the sample complexity of unlabeled examples in this setting, and we believe it constitutes a meaningful contribution to the literature. We would also like to highlight that Theorem 3 was among the most technically challenging results in the paper.
> - **Clarity of assumptions:** Any additional assumptions required for a given theorem are stated explicitly within that theorem. Otherwise, the results hold for general concept classes. Let us also briefly summarize every major assumption made in the paper (by major we are not including obvious ones such as bounded VC dimension, etc.). Note that all of them appear in Section 4. For the SAR setup (Theorem 8), we assumed a lower bound on the weight ratio, and we motivated its necessity in lines 207–212. For the PCS setup, in addition to the lower bound on the weight ratio, we assumed that the domain is $[0, 1]^k$ and imposed a $\gamma$-margin assumption. We demonstrated the necessity of the $\gamma$-margin assumption in Theorem 10. We will add further discussion of these assumptions and mark them more explicitly in the text.
> - **Relation to one-sided error:** This is a good observation. First, note  that for any intersection-closed concept class with finite VC dimension, we have $C_\cap = C$, so the assumption $\operatorname{VCD}(C_\cap) < \infty$ is indeed weaker than assuming both intersection-closure and finite VC dimension.
> There are indeed connections between realizable PU learning without access to unlabeled examples (i.e., learning from positive-only examples) and learning with one-sided error. In short, learning with one-sided error corresponds to realizable learning from positive-only examples  while imposing two additional constraints: (i)  the learner must be proper, and (ii) the false negative must be zero. Consequently,  stronger assumptions are required for one-sided error learning. We will clarify this connection further in the camera-ready version.
> - **Relationship between claw number and other combinatorial parameters:**  We discussed the relationship between $\operatorname{VCD}(C_\cap)$ and the claw number in Proposition 4. In fact, $\operatorname{VCD}(C_\cap)$ was the original motivation for defining the claw number. However, at present, we are not aware of any connection between the claw number and other established combinatorial parameters beyond VC dimension. We agree that this is an excellent direction for future research.
> - **Lower bounds on the complexity of unlabeled examples:**  One fundamental difference between PU learning and semi-supervised learning is that, except for certain restricted classes, learning from positive-only examples is generally not possible. This makes it less surprising that we are able to establish lower bounds on the number of unlabeled examples required. However, an interesting line for future work is  “For which setups we can elevate the need for positive examples”. Theorem 1 in our paper shows that reducing the number of required labeled examples necessitates additional assumptions either on the marginal distribution or the concept class (as is  the case in semi-supervised learning.)
>
> We also thank the reviewer for pointing out those issues, and we will address them in the camera-ready version.

---

> ### Comment · Reviewer_8f7e · 2025-08-03
>
> Rereading the paper I believe there is a tiny issue with Def. 2 (the claw number). Independently of $\frak h$ (even if its 0) the definition requires sets $B$ of arbitrary large size $m\ge \frak h$. However if the domain is finite this is not possible. I guess a fix like $|X|\ge m \ge \frak h$ is sufficient?
>
> *One small clarification on one-sided*: I did not meant to say that the class is intersection-closed and has finite VC dimension. I meant to say the "intersection closure" of a class $C$ (i.e., what you call $C_\cap$) has finite VC dimension (see, e.g., Kivinen 1995). So this is exactly as in your Proposition 4. I digged a bit deeper, and in fact there seems to be some relationship between claw number and Kivinen's slicing dimension. E.g., (haven't though deeply here) for $\frak h=1$ you get sets $B$, which are "sliced", of arbitrary size. Thus the slicing dimension is infinite. Also a similar fact is also captured by the fact that the VC-dim of $C_\cap$ is the (one-centered) star number of the original class $C$ (see Theorem of 19 of Hanneke 2024). Having said that it is perhaps better to (at least partially) attribute Proposition 4 to these 2 papers, if what I said is valid.
>
> Kivinen, Jyrki. "Learning reliably and with one-sided error." Mathematical systems theory 28.2 (1995): 141-172.
> Hanneke, Steve. "The star number and eluder dimension: Elementary observations about the dimensions of disagreement." COLT 2024.

---

> ### Author Response · Authors · 2025-08-05
> **Official Comment by Authors**
>
> We would first like to thank the reviewer for pointing out these papers. We were not aware of Kivinen (1995). You are correct that both the slicing dimension and the one-centered star number are equal to $VCD(C_\cap)$. While we had observed that $VCD(C_\cap)$ lower bounds the star number, we were not aware that the notion of the one-centered star number has been explicitly discussed in the literature. We will include a discussion of this relationship in our revised manuscript.
>
>
> ### Clarification on learning from one-sided error:
> The definitions of learning from one-sided error in Kivinen (1995) and Natarajan (1987) differ slightly. Both require the learner to have zero false negative. However, Natarajan (1987) additionally assumes that the training data consists solely of positive examples and that the learner is proper.  Specifically, Natarajan (1987) defines one-sided error as follows:
>
> > Following (Valiant, 1984), we say that a family of functions $F$ is learnable with one-sided error if and only if there exists an algorithm that
> > (a) makes polynomially many calls of EXAMPLE both in an adjustable error parameter $h$ and in the number of variables $n$. The algorithm need not run in polynomial time.  \
> > (b) For all functions $f \in F$, and all probability distributions $P$ over the assignments, the algorithm deduces with probability $(1 - 1/h)$ **a function $g \in F$**, such that for any assignment $a$:
> > $$ g(a)=1 \implies f(a)=1$$
> > $$ \text{if } S=\{a \mid f(a)=1 \text{ and } g(a)=0\}, $$
> > $$ \text{ then } \quad \sum_{a \in S} P(a) \leq \frac{1}{h}. $$
>
>
> The condition $g \in F$ enforces properness. They also define a learning algorithm as follows:
>
> > A learning algorithm is an algorithm that attempts to infer a function $f$ on the variables, from positive examples for $f$.
>
> This shows that they require training data to be positively labeled. By contrast, in Definition 4.1 of Kivinen (1995) I cannot find any such restrictions. In Section 2.4, they explicitly emphasize that they do not impose properness:
>
> > In PAC learning it is sometimes required that the hypothesis must be from the concept class to be learned. In reliable learning and learning with one-sided error the criteria for a good hypothesis are tighter, and we must allow the learner to choose its hypothesis from a larger set. In learning with one-sided error we accept as a hypothesis any measurable set.
>
>
> ### Natarajan (1987) condition:
> The characterizing condition that Natarajan (1987) establishes for their setup in Theorem 3 is that $F$ has **polynomial dimension**. Having polynomial dimension entails that the concept class is **well-ordered**, i.e., the intersection of every two concepts remains in the class (we referred to this property as intersection closedness in our rebuttal).
>
>
> ### On loosening the $|X| = \infty$ condition in the definition of claw number :
> A fixed $m$ is insufficient in the definition of claw number. This is because any concept class over a finite domain $X$ satisfies $VCD(C_\cap) \leq |X| < \infty$. By the results of Lee et al. (2025), such classes can be learned with zero unlabeled examples, and it is impossible to establish a lower bound on unlabeled examples. Consequently, the notion of claw number should be zero when the domain has a finite size.
>
> ### References
> Lee, J. H., Mehrotra, A., and Zampetakis, M. (2025). Learning with positive and imperfect unlabeled data. arXiv preprint arXiv:2504.10428.

---

> > ### Comment · Reviewer_8f7e · 2025-08-05
> >
> > Thanks! I will raise my score.

---

> > > ### Author Response · Authors · 2025-08-05
> > > **Official Comment by Authors**
> > >
> > > Thank you. We truly appreciate it.

---

### Official Review · Reviewer_cR8L · 2025-07-01

**Clarity:** 2
**Significance:** 3
**Originality:** 2
**Rating:** 4
**Confidence:** 3

**Summary:**

This paper provides a theoretical analysis of PU learning. First, it identifies different PU learning settings, including cases with covariate shift of positive data. Then, it provides a detailed analysis of the sample complexity of these settings. The paper offers new insights into understanding PU learning with distribution shifts.

**Questions:**

Please see "weaknesses".

**Ethical Concerns:**

["NO or VERY MINOR ethics concerns only"]

**Final Justification:**

Thanks for the rebuttal. I will maintain my score.

**Limitations:**

The limitations should be discussed.

**Paper Formatting Concerns:**

None.

**Quality:**

2

**Strengths And Weaknesses:**

## Strengths
- The theoretical analysis is comprehensive and well-executed.

- The studied problem, i.e., the sample complexity of different PU learning settings, is important to the literature.

- The presented theoretical results are interesting and novel, as shown in Table 1.

## Weaknesses

- The paper is novel for the PU learning literature. However, most of the analysis seems dependent on Ben-David and Urner (2012). Therefore, the authors should justify how their proposed setting differs significantly from that of Ben-David and Urner (2012). Additionally, could the authors discuss the differences with [1]?

- The theoretical analysis of the paper is good and comprehensive. The study of different PU learning settings is also valid and sound. However, I am not familiar with the sample complexity analysis of transfer learning in Ben-David and Urner (2012), so it may be difficult for me to justify the soundness of the proposed theory.

- Since the paper contains many theoretical analyses, it is important to write them in a clearer way so that readers can understand them.

- The proposed assumptions need further discussion and justification. Additionally, it is unclear how the theoretical analysis can be applied to real-world applications. There are no experimental results in the paper.

[1] Coudray O, Keribin C, Massart P, et al. Risk bounds for positive-unlabeled learning under the selected at random assumption[J]. Journal of Machine Learning Research, 2023, 24(107): 1-31.

---

> ### Author Rebuttal · Authors · 2025-07-31
>
> Thank you for your constructive feedback and helpful suggestions. We would like to address your questions and concerns bellow:
> - **Dependence on Ben-David and Urner (2012):** While we clearly stated that our results in Section 4 are inspired by Ben-David and Urner (2012), our application  to PU learning and the results in this domain are new. Furthermore, as described in lines 265–270, the application to PU required significant alterations to Ben-David and Urner (2012). Also, note that Section 4 constitutes only a small portion of our overall contribution.
> - **Comparison with Coudray et al (2023):** The results of Coudray et al. (2023) are for the SAR setup; however, they are not directly comparable to ours. Their work addresses the more general agnostic PU learning setting and assumes knowledge of the class prior $\alpha$, whereas we focus on the realizable setting without requiring knowledge of $\alpha$.
> - **Further discussion about assumptions:** Let us briefly summarize every major assumption made in the paper (by major we are not including obvious ones such as bounded VC dimension, etc.). Note that all of them appear in Section 4. For the SAR setup (Theorem 8), we assumed a lower bound on the weight ratio, and we motivated its necessity in lines 207–212. For the PCS setup, in addition to the lower bound on the weight ratio, we assumed that the domain is $[0, 1]^k$ and imposed a $\gamma$-margin assumption. We demonstrated the necessity of the $\gamma$-margin assumption in Theorem 10. We will add further discussion of these assumptions and mark them more explicitly in the text.
> - **Lack of experiments:** The *NeurIPS* conference welcomes a broad range of contributions, including purely theoretical work. Our submission falls within this scope.

---

> > ### Comment · Reviewer_cR8L · 2025-08-05
> >
> > Thanks for the rebuttal. I will maintain my score.

---

### Official Review · Reviewer_13uY · 2025-07-02

**Clarity:** 3
**Significance:** 3
**Originality:** 3
**Rating:** 4
**Confidence:** 3

**Summary:**

This paper presents a comprehensive theoretical analysis of the statistical complexity of PU learning. In contrast to most prior work, this study does not require the class prior, α, to be known by the learner. The paper explores both the realizable and agnostic learning paradigms across four distinct data-generating settings: Selected Completely at Random (SCAR), Selected at Random (SAR), Positive Covariate Shift (PCS), and Arbitrary Positive Distribution Shift (APDS) .The main contributions include: 1. Providing nearly tight upper and lower sample complexity bounds for the SCAR and SAR settings. 2. Establishing the first finite-sample generalization guarantees for the PCS and APDS settings.3. Introducing a novel combinatorial parameter, the "claw number," to characterize the lower bound on unlabeled sample complexity.
4. In the agnostic setting with an unknown α, proving that a learner can achieve an error bound proportional to the optimal error and showing that this proportionality factor is theoretically tight.

**Questions:**

The exponential dependency on dimensionality in your bound for Algorithm 1 seems to be a significant practical barrier. Could you comment on the tightness of this bound? Are there avenues for improvement, or could you outline specific scenarios where this complexity would be manageable?

**Ethical Concerns:**

["NO or VERY MINOR ethics concerns only"]

**Final Justification:**

The authors have successfully adressed the majority of my concerns as mentioned in the rebuttal.

**Limitations:**

As noted, the results for challenging distribution shifts rely on a margin assumption, which limits the applicability of these specific conclusions.

**Quality:**

3

**Strengths And Weaknesses:**

Strengths:
1. The most significant contribution of this work is its systematic treatment of the unknown class prior setting. This addresses a long-standing bottleneck in PU learning theory and brings the theoretical analysis closer to real-world application scenarios.
2. The paper is impressively comprehensive, systematically analyzing settings from the well-studied SCAR to the more challenging PCS and APDS. By providing both upper bounds (algorithmic guarantees) and lower bounds (inherent problem difficulty) for several key settings, the analysis is remarkably complete and convincing.
3. The introduction of the "claw number" is an interesting conceptual innovation. Furthermore, providing the first finite-sample guarantees for the PCS and APDS settings, where such results were previously lacking, represents a significant theoretical breakthrough.

Weaknesses:
1. To achieve theoretical guarantees in more challenging settings like PCS, the paper introduces new, strong assumptions, such as the γ-margin assumption. This leads to a sample complexity for unlabeled data that is exponential in the data dimensionality.
2. This is a purely theoretical paper with no experimental section. While the theoretical contributions are self-contained and sufficient, the absence of experiments on synthetic or real-world datasets makes it difficult to assess the practical performance of the proposed algorithms (e.g., Algorithm 1), their sensitivity to hyperparameters, and the tightness of the theoretical bounds in practice.

---

> ### Author Rebuttal · Authors · 2025-07-31
>
> We would like to thank the reviewer for assessing our work and for valuable feedback.  The reviewer expressed concern about $\gamma$-margin assumption and sample complexity for unlabeled data that being exponential in the data dimensionality: Informally, Theorem 10 demonstrates that in PCS setup without $\gamma$-margin assumption, any PU learner that achieves error less than $\varepsilon$ with probability more than $1 - \delta$ should at least receive a total number of positive and unlabeled examples which is in order of square root of domain size. Consequently, the task of PU learning in the PCS setup without $\gamma$-margin assumption is impossible. Similarly, Theorem 11 demonstrates that the exponential dependence of the total number of unlabeled and positive examples is unavoidable in the PCS setup.  Interestingly, our results for the SAR setup (Theorem 8) do not require either of these assumptions. This highlights that the PCS setup is inherently more challenging than the SAR setup.

---

### Official Review · Reviewer_oWVS · 2025-07-03

**Clarity:** 3
**Significance:** 3
**Originality:** 3
**Rating:** 4
**Confidence:** 3

**Summary:**

This study revisits a basic issue in PU learning. The authors define PU learnability and discuss under which conditions we can obtain a desirable classifier. Their analysis investigates each setting in PU learning, including those considered by Elkan and Noto (2008) and du Plessis et al. (2015), and provides a comprehensive framework.

**Questions:**

See above.

**Ethical Concerns:**

["NO or VERY MINOR ethics concerns only"]

**Final Justification:**

As a result of the discussion, I feel that the contribution has become clearer. While the study is interesting, the results are not particularly surprising, so I will give it a weak accept.

**Limitations:**

N/A.

**Paper Formatting Concerns:**

None.

**Quality:**

3

**Strengths And Weaknesses:**

The authors address a fundamental problem in PU learning. Although their problem setup is interesting, I have several questions regarding their results. I list the points below:
- I believe the problem setup in Elkan and Noto (2008) and its subsequent works differs from that in du Plessis et al. (2015) and related studies. The former is referred to as censoring PU learning, while the latter is known as case-control PU learning. The difference is not related to knowledge of the class prior. Even when the class prior is known, the data-generating processes are distinct. It seems that the authors' setting largely follows that of du Plessis et al. (2015). If they were to follow the setting in Elkan and Noto (2008), we would usually assume access only to a single dataset from which labels can be observed. For an explanation of this difference, see Niu et al. (2016).
- Theorem 14 appears to be somewhat trivial. The Bayes classifier is usually expressed as $p(y = 1 \mid x) = \alpha, p(x \mid y = 1) / p(x)$, where $p(x \mid y = 1)$ is the class-conditional density of $x$, $p(x)$ is the marginal density, and $\alpha$ is the class prior. This density ratio $p(x \mid y = 1) / p(x)$ can be estimated via density-ratio estimation methods. Consequently, it is rather obvious that we can obtain the result stated in Theorem 14, because we can recover the ordering of $p(y = 1 \mid x)$; the exact classification boundary is then determined by the class prior. For the relationship between density-ratio estimation and PU learning, see Kato and Teshima (2021).

Reference:
- Niu, du Plessis, Sakai, Ma, and Sugiyama (2016). Theoretical Comparisons of Positive-Unlabeled Learning against Positive-Negative Learning. NeurIPS.
- Kato and Teshima (2021). Non-Negative Bregman Divergence Minimization for Deep Direct Density Ratio Estimation. ICML.

---

> ### Author Rebuttal · Authors · 2025-07-31
>
> Thank you for carefully evaluating our work and for offering constructive feedback. We would like to address your questions bellow:
> 1. **Case-control vs Censoring PU learning:** Yes, our paper focuses on the case-control PU learning setting. We tried to emphasize this on line 40 of the manuscript (although we mistakenly referred to it as "control set PU learning" – this was a typo). While we briefly outlined the distinction between censoring PU learning and case-control PU learning in lines 34-37, we will happily explain this distinction in more detail. We followed the naming convention from the survey by Bekker and Davis (2020), which is why we cited their work. We will update the citations to Elkan and Noto (2008) and du Plessis et al. (2015).
> 2. **Relation with Kato and Teshima (2021):** We agree that there is a connection between density ratio estimation and PU learning. However, the results presented in Section 5.2 of our paper hold for any hypothesis class with VC dimension $d$ and any distribution with class prior $\alpha$. In contrast, Theorem 1 of Kato and Teshima (2021) relies on strong Assumptions 3 and 4 introduced in Section 4.1, and does not provide any lower bound. In our Theorem 14, we show that our upper bound on the generalization error is tight. \
> The difference between our work and that of Kato and Teshima (2021) is further highlighted by the fact that Theorem 1 in their paper shows that the error of the BD minimization method can approach the error of the best concept in the class as the number of training examples goes to infinity. In contrast, our Theorem 14 demonstrates that, for a given $\alpha$, *no* algorithm can achieve an error better than a factor of $\frac{\max(\alpha, 1 - \alpha)}{\min(\alpha, 1 - \alpha)}$ times the error of the best classifier in the class regardless of  the number of unlabeled or positive training examples. \
> We will include an extended discussion clarifying the relationship between our work and that of Kato and Teshima (2021).
>
> References: \
> Bekker, Jessa, and Jesse Davis. "Learning from positive and unlabeled data: A survey." Machine learning 109.4 (2020): 719-760.

---

> > ### Comment · Reviewer_oWVS · 2025-08-05
> > **Reply to the authors**
> >
> > Thank you for your reply.
> >
> > 1. I understand the point. Thank you for the clarification.
> >
> > 2. We also appreciate the authors' explanation. For finite-sample analysis, I believe Zheng et al. (2023) may be helpful for your study. They provide non-asymptotic error bounds for density-ratio estimation. As shown by Kato and Teshima, in the context of Bregman divergence for density-ratio estimation, PU learning losses can be applied. Intuitively, I think these strands of literature could be connected to your study.
> >
> > - Zheng, Shen, Jiao, Lin, and Huang (2023). "An Error Analysis of Deep Density-Ratio Estimation with Bregman Divergence."

---

> > > ### Author Response · Authors · 2025-08-07
> > > **Official Comment by Authors**
> > >
> > > We thank the reviewer for their valuable reference. As stated in our original rebuttal, the results of Keto and Teshima are conditioned upon several strong assumptions. Similarly, the results of Zheng et al rely on imposing restricting assumptions on the relevant data distributions (Assumption 1, 2, 3 in Section 3 and Assumption 4 in Section 4). Our results address the setup where no such assumptions are assumed. Furthermore, as mentioned in our original rebuttal, our Theorem 14 implies that the type of convergence to the class approximation error claimed in Kato and Teshima's results is not achievable without additional assumptions. Similarly, Theorem 1 of Zheng et al shows that the error converges to a constant multiple of the approximation error, which, due to our Theorem 14, would not be possible without additional assumptions. In the next version of our submission we will cite these papers and explicitly explain the differences between their setups and our results.

---

> > > > ### Comment · Reviewer_oWVS · 2025-08-08
> > > > **Reply to the authors**
> > > >
> > > > Thank you for your detailed comments! The authors have addressed my concerns. I understood the contributions and will maintain my original score.

---

### Note · Authors · 2025-08-15

We sincerely thank all reviewers for their efforts in improving this paper. During the rebuttal phase, we addressed each question raised by the reviewers, and in response to their suggestions, we will implement the following revisions:
- We will expand the Related Work and Contribution sections to better highlight the distinctions between our work and existing literature. In particular, we will explicitly discuss the assumptions required by the existing results on density-ratio estimation [Zheng et al. (2023), Kato and Teshima (2021)], assumptions that we do not make.
- Theorem 1.3 of Lee et al. (2025) showed that no concept class $C$ with $VCD(C_\cap) = \infty$ can be learned using only positive examples. In Proposition 4, we observed that the claw number is greater than zero if and only if $VCD(C_\cap) = \infty$. In contrast with Theorem 1.3 of Lee et al, the setup we consider also allows unlabeled training examples. Our results (combining Theorem 3 and proposition 4) show that not only positive examples alone do not suffice when  $VCD(C_\cap) = \infty$ but there is a concrete lower bound on the number of required unlabeled examples for distribution-free learning (as a function of the claw number). Reviewer 8f7e further noted that the notion of $VCD(C_\cap)$ has been studied in other contexts [Kivinen (1995), Hanneke (2024)], and we will clarify this connection.
- We will further clarify the assumption we make in the paper and explain their necessity.
- We will further clarify the distinction between the setups of Case-control and Censoring PU learning in the Introduction.


References:
- Zheng, Shen, Jiao, Lin, and Huang (2023). "An Error Analysis of Deep Density-Ratio Estimation with Bregman Divergence."
- Kato and Teshima (2021). Non-Negative Bregman Divergence Minimization for Deep Direct Density Ratio Estimation. ICML.
- Lee, J. H., Mehrotra, A., and Zampetakis, M. (2025). Learning with positive and imperfect unlabeled data. arXiv preprint arXiv:2504.10428.
- Kivinen, Jyrki. "Learning reliably and with one-sided error." Mathematical systems theory 28.2 (1995): 141-172.
- Hanneke, Steve. "The star number and eluder dimension: Elementary observations about the dimensions of disagreement." COLT 2024.

---

### Decision · Program_Chairs · 2025-09-17

**Decision:**

Accept (poster)

**Comment:**

This paper presents new sample complexities for positive and unlabeled learning based on relaxed assumptions and four problem setups.  In particular, it presents upper and lower bounds on the required training sample sizes, for both positively labeled and unlabeled data, without assuming class prior is known to the learner (which is assumed in most prior studies). It also introduces a novel quantity called claw number to characterize lower bounds.

---

Overall, the work is solid. All reviewers are on the positive side. Many reviewers compliment on the comprehensiveness of setups.

Most reviewers request more thorough comparisons with related works. Authors did a plausible job in responses. I should mention the concept of claw number is also created in graph theory [1]. Authors are encouraged to discuss potential connections to it and give proper credit to it.

[1] Aditya and Chandran, Cubicity of interval graphs and the claw number.

Reviewers also provide many good suggestions on improving the clarity of presented theorems and providing more thorough discussions on them. Authors are encouraged to incorporate them in revision.